# Contrastive Novelty Learning: Anticipating Outliers with Large Language Models

## Abstract

In many task settings, text classification models are likely to encounter examples from novel classes on which they cannot predict correctly. Selective prediction, in which models abstain on low-confidence examples, provides a possible solution, but existing models are often overly confident on OOD examples. To remedy this overconfidence, we introduce Contrastive Novelty Learning (CNL), a two-step method that generates OOD examples representative of novel classes, then trains to decrease confidence on them. First, we generate OOD examples by prompting a large language model twice: we prompt it to enumerate novel classes relevant to the label set, then generate examples from each novel class matching the task format. Second, we train our classifier with a novel contrastive objective that encourages lower confidence on generated OOD examples than training examples. When trained with CNL, classifiers improve in their ability to detect and abstain on OOD examples over prior methods by an average of 2.3% AUAC and 5.5% AUROC across 4 NLP datasets, with no cost to in-distribution accuracy.[1]

## 1 Introduction

Recent progress in NLP has led to text classification models that are accurate not only in-distribution, but also on some out-of-domain data (Arora et al., 2021). Nonetheless, some categories of real-world distribution shift still pose serious challenges. For instance, in open-set label shift, the test data includes examples from novel classes not present in the training data, making it impossible for a standard classifier to predict correctly (Scheirer et al., 2013). Moreover, novel class examples can be difficult to detect with conventional OOD detection methods, as they typically bear a strong surface resemblance to training examples (Țifrea et al., 2021). In this paper, we frame open-set label shift as a selective prediction problem (El-Yaniv & Wiener, 2010; Geifman & El-Yaniv, 2017) that we call open-set selective classification (OSSC). OSSC requires text classifiers to predict correctly on closed-set examples while abstaining on novel class examples.

To perform well on OSSC, a classifier must have lower confidence on novel class examples than closed-set examples by learning features which differentiate novel classes from closed-set classes (Perera et al., 2020). In order to supervise this representation learning, it is useful to identify what examples from novel classes might look like. Prior work has explored automatically generating OOD images by adding random perturbations to ID examples (Setlur et al., 2022). Text inputs, however, are composed of discrete tokens, and modifying even a single token can unpredictably alter the meaning of a sentence. We seek an automatic generation method that addresses these limitations, leveraging the generative ability of large language models (LLMs) like GPT-3 (Brown et al., 2020). LLMs are a desirable source for novelty, as their generation is informed by a broad corpus of examples seen during pretraining, allowing them to reliably generate from classes outside a dataset.

We present Contrastive Novelty Learning (CNL), a method to improve the OSSC ability of a classifier by automatically generating OOD examples and then training to abstain on them. To generate a diverse set of OOD examples that anticipate different potential test-time shifts, we introduce Novelty Prompting, a method that augments a source dataset with novel class examples generated by a LLM. We first perform label generation, prompting our LLM to extend the closed-set labels with novel labels. We then prompt the LLM to generate new examples conditioned on each novel label

---

[1]Code and data have been uploaded and will be released.

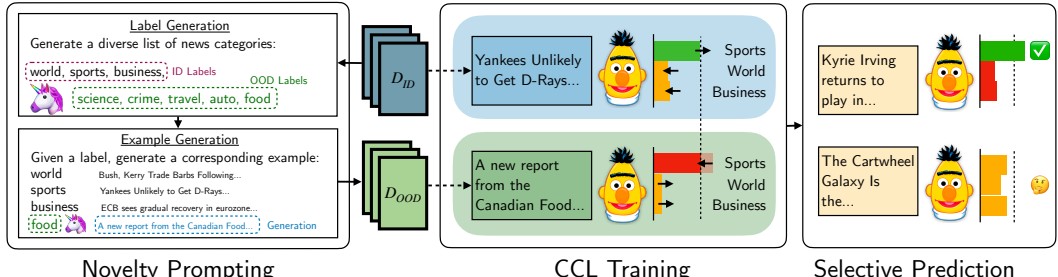

Figure 1: *Contrastive Novelty Learning pipeline.* We Novelty Prompt a generator model to produce a novel set $D_{\text{OOD}}$, then train with a contrastive confidence loss (CCL) on our original train set $D_{\text{ID}}$ and $D_{\text{OOD}}$, ensuring that our classifier is *less confident* on generated novel examples than closed-set examples. Finally, we abstain based on model logits.

to form a large set of probable novel examples. Finally, we propose a contrastive confidence loss (CCL) for training, which encourages both high accuracy on the ID training set and lower relative confidence on the generated novel examples. We show that CCL outperforms stricter losses like Outlier Exposure (Hendrycks et al., 2019), which can adversely affect ID performance. Our full pipeline is shown in Figure 1. Our method can be viewed as a form of "partial" knowledge distillation: we leverage an LLM "teacher model" to improve novelty detection performance without altering the student model's strong ID classification ability.

We evaluate CNL against state-of-the-art baselines across 14 splits of 4 datasets (AGNews (Zhang et al., 2015), TREC-10 (Li & Roth, 2002), TACRED (Zhang et al., 2017), Emotion (Saravia et al., 2018)), finding that it improves both OOD detection and OSSC, by an average of 5.5% AUROC and 2.3% AUAC over the best prior method. These improvements come at no cost to ID accuracy, demonstrating that it is possible to distill novelty detection alone without affecting predictive power. Finally, we analyze the settings in which CNL can improve OSSC performance. In the data dimension, scale is often optional: with as few as 1000 generated examples, we find an improvement over vanilla training on all 4 datasets. The same is only partially true for LLM size, as on some datasets only a sufficiently large model can generate useful examples.

## 2 PROBLEM SETTING

### 2.1 OPEN-SET SELECTIVE CLASSIFICATION

In standard classification, an optimal model $f$ should predict the ground-truth label $y$ of an input example $x$ from a closed set of *known* labels $\mathcal{Y}_{\text{ID}}$. However, under a more realistic *open-set* setting, some test examples are drawn from *unknown* novel classes $\mathcal{Y}_{\text{OOD}}$. Without a priori knowledge of $\mathcal{Y}_{\text{OOD}}$, a standard discriminative classifier will never correctly classify a novel example. Instead, an optimal open-set selective classifier $f$ should predict $y$ when $y \in \mathcal{Y}_{\text{ID}}$, and abstain otherwise.

For a probabilistic model $p_\theta(y \mid x)$ and associated confidence metric, the corresponding prediction is given by $f(x) = (\hat{y}, c)$, where $\hat{y} = \arg\max_{y \in \mathcal{Y}_{\text{ID}}} p_\theta(y \mid x)$ and $c$ denotes the model's confidence. When used as a selective classifier with threshold $\gamma$, $f$ predicts $\hat{y}$ when $c > \gamma$ and abstains otherwise (Geifman & El-Yaniv, 2017). This differs from OOD detection (Hendrycks & Gimpel, 2017) in that $f$ must abstain on both novel examples and its own errors and must attain high ID accuracy.

### 2.2 EVALUATION PROTOCOL

We holistically measure selective classification performance with the area under the accuracy-coverage curve (AUAC). The accuracy-coverage curve plots accuracy as a function of the fraction of examples on which the model predicts (i.e., coverage) as the confidence threshold $\gamma$ varies. For accuracy computation, we treat predictions on all novel class examples as incorrect. AUAC measures the combined ability of a model in ID classification accuracy, ID calibration, and OOD detection.

Though we deviate from prior work in reporting AUAC, to demonstrate that our method is still effective in OOD detection, we additionally compute the Area under the ROC curve (AUROC). AUROC measures a model's ability to detect when a test example $x$ is of a novel class ($y \in \mathcal{Y}_{\text{OOD}}$). Higher is better: 50% AUROC is random, and 100% is perfect.

## 3 METHOD: CONTRASTIVE NOVELTY LEARNING

Here we describe Contrastive Novelty Learning, a method for automatically improving OSSC. At a high level, we generate novel examples and then train to ensure that our model is *less confident* on generated novel examples than closed-set examples. We first describe desiderata for useful novelty, then introduce a two-phased novel example generation method, Novelty Prompting, and finally introduce a contrastive confidence loss for classifier training. We illustrate the method in Figure 1.

### 3.1 NOVELTY PROMPTING

**Desiderata of Novelty Generation** Inspired by previous work which utilize known, representative OOD data to train selective prediction and OOD detection models (Kamath et al., 2020; Hendrycks et al., 2019), we focus on creating an generated "novel set" that is representative of potential label shifts at test time. The "novel set" must be (1) *plausible*, meaning that it should bear a surface resemblance to the training data, e.g., we should create news examples for a news dataset, and (2) *semantically novel*, meaning that these examples should be from new classes. For example, selecting data from an entirely separate dataset, as is done in Hendrycks et al. (2019), violates plausibility. Meanwhile simply editing surface features or recombining examples as is done in mixup (Zhang et al., 2018) might induce a distribution shift but would not result in semantic novelty.

To satisfy these desiderata, we propose a two-stage generation method called Novelty Prompting (NP). To encourage semantic novelty, we first generate novel labels given a dataset's extant labels. We then show existing examples to a language model (to encourage plausibility) and ask it to generate a new example conditioned on one of the new labels. Figure 1 shows both prompt formats.

**Label Generation** Though prompting with large autoregressive language models (LLMs) like GPT-3 has typically been explored in the context of few and zero-shot learning to perform standard NLP tasks Brown et al. (2020), we find that LLMs are also capable of "expanding" a set of topically related concepts that might realistically co-occur via sequence continuation.

We leverage this capability to generate novel labels. We prompt the largest GPT-3 model available (Davinci) with a task-specific instruction as well as the concatenation of the normalized known ($\mathcal{Y}_{\text{ID}}$) labels. [2] By taking the union over continuations of one or more novel labels $N$ times, we obtain a diverse "novel label set." We combine multiple completions because in preliminary experiments, we observed that single completions tend to overgenerate labels from a narrow subcategory of possible novel classes. To remedy concerns about data leakage due to dataset examples of the true unknown class possibly appearing in LLM pretraining, we remove instances of the gold novel label(s) from this set. In practice, predicting the true novel test-time labels is both permissible and desirable, so our experimental setup likely underestimates our method's performance.

Finally, we filter out generated labels that are closely related to ID labels. For example, if `joy` appears in the ID labels, we remove synonyms like `happiness`. We use a large online thesaurus[3] to identify synonyms from the final novel label set. We analyze the impact of filtering in Appendix A.8.

**Example Generation** To generate each novel example, we randomly sample a novel label from our set and prompt a LLM (we use GPT-J) to generate an example of that label. We prime this model with one random sampled label-example pair from each ID class in the training dataset in the prompt, resulting in 3-6 in-context examples, varying based on the dataset. Providing these context pairs ensures that our generation is plausible: the model is encouraged to generate a specific style of text. We perform this generation procedure repeatedly to form a novel example set. We show the prompt we use for this step in Appendix A.1, and several generated label-example pairs in Figure 2.

---

[2]Though this requires some human intervention, it both (1) satisfies the true zero-shot nature of test-time label shift as it requires no knowledge of the unknown labels and (2) requires minimal effort, typically only involving converting an abbreviation label such as `LOC` into `Location`.

[3]https://moby-thesaurus.org/

## 3.2 Contrastive Confidence Loss Training

Our second contribution is an improved loss function for training models to have lower confidence on OOD examples than ID examples. Prior work has used the Outlier Exposure (Hendrycks et al., 2019) objective, which encourages the model $f$ to output a uniform probability distribution over closed-set classes when given a novel example $x$. Hendrycks et al. (2019) successfully used Outlier Exposure to train models on OOD data gathered from a different dataset (e.g., Wikitext), since there was very little risk of this data overlapping with ID data. In contrast, we automatically generate plausible novel examples, which runs the risk that some of our novel examples will inevitably be in distribution. Since Outlier Exposure encourages models to have the lowest possible confidence on novel examples, it can hurt predictive accuracy when some examples $x$ actually resemble closed-set examples. Instead, we seek a solution which treats outliers more flexibly.

We propose a novel contrastive confidence loss (CCL) that encourages models to be less confident on OOD examples than ID examples. This is a less strict objective as models can achieve minimum loss *without* predicting a perfectly uniform distribution for the generated novel examples. For an input $x$, let $p_\theta(y \mid x)$ be the model's predicted distribution over $\mathcal{Y}_{\text{ID}}$. Let $c_\theta(x) = \max_{y \in \mathcal{Y}_{\text{ID}}} p_\theta(y \mid x)$, the Maximum Softmax Probability (MaxProb) (Hendrycks & Gimpel, 2017), which we use as our confidence metric. Finally, let $\ell$ denote the cross-entropy loss with a one-hot target vector, and $D_{\text{ID}}$ and $D_{\text{OOD}}$ denote the training set and novel set respectively. We define CCL as follows:

$$\mathcal{L}(\theta) = \mathbb{E}_{(x_{\text{id}}, y_{\text{id}}) \sim D_{\text{ID}}}[\ell(p_\theta(y \mid x_{\text{id}}), y_{\text{id}})] + \lambda \mathbb{E}_{x_{\text{id}} \sim D_{\text{ID}}, x_{\text{ood}} \sim D_{\text{OOD}}}[\max(0, c_\theta(x_{\text{ood}}) - c_\theta(x_{\text{id}}))]. \quad (1)$$

That is, we penalize the confidence of novel examples which have higher confidence than any closed-set example. While this still induces our model to learn lower confidence on novel examples, it simultaneously permits our model to learn that *some novel examples can be more novel than others*, rather than learn minimal confidence on *all* members of the generated novel set. In practice, we obtain an unbiased estimate of the second term by sampling a batch of $n$ ID and $n$ OOD examples at each step and computing the second term pairwise between each of the $n^2$ ID-OOD example pairs. We arbitrarily choose $\lambda = 1.0$, weighting the two terms of the objective equally.

## 4 Experimental Setup

### 4.1 Datasets

We construct artificial dataset splits from 4 popular NLP classification datasets by holding out one or more labels from training and moving all examples of that label to the test split, removing classes that are too small to yield statistical significance in our evaluations. Specifically, we evaluate a question intent detection dataset, TREC-10 (Li & Roth, 2002) and construct 5 splits. We also evaluate two popular topic classification datasets, AGNews (Zhang et al., 2015), a news classification dataset, and Emotion (Saravia et al., 2018), a tweet classification dataset. We construct 4 splits for each. Finally, we evaluate TACRED (Zhang et al., 2017), a strongly class-imbalanced sentence relation-classification dataset with a large set of 41 possible relations, for which we construct a single split where we hold out the 35 smallest classes. Further dataset details are outlined in Appendix A.6. Results for each dataset are averaged across all splits.

### 4.2 Experimental Details

For Novelty Prompting, we perform label generation using the best available GPT-3 model, GPT-3 Davinci (Brown et al., 2020) and example generation with a smaller pretrained GPT-J 6B model Komatsuzaki (2021). To generate a sufficiently diverse novelty set, we perform 5 label generation iterations, then generate 100000 examples. We analyze the impact of generation budget in Section 5.3. We train BERT-base classifiers with CCL for 5000 steps and batch size $n = 40$. On TACRED, we permit only generations containing exactly two entities, one of which is a subject and the other an object, filtering out roughly 90% of generations. This is a task constraint: the relation between any other number of entities is under- or overspecified. We describe further details in Appendix A.7.

| AUAC ($\uparrow$) | TREC-10 | AGNews | Emotion | TACRED | Average |
|---|---|---|---|---|---|
| Vanilla | $89.2_{\pm 2.2}$ | $87.9_{\pm 0.6}$ | $90.3_{\pm 1.0}$ | $89.6_{\pm 0.1}$ | 89.3 |
| kFolden | $\mathbf{93.5}_{\pm 0.6}$ | $85.8_{\pm 1.6}$ | $90.6_{\pm 0.9}$ | $84.9_{\pm 3.5}$ | 88.7 |
| Contrastive | $92.0_{\pm 0.4}$ | $87.0_{\pm 0.9}$ | $92.2_{\pm 0.4}$ | $88.8_{\pm 0.7}$ | 90.0 |
| CCL + Wikitext | $91.2_{\pm 1.4}$ | $88.6_{\pm 0.6}$ | $92.0_{\pm 0.4}$ | $89.3_{\pm 0.5}$ | 90.3 |
| CCL + Zero-Shot | $92.5_{\pm 0.8}$ | $89.1_{\pm 0.4}$ | $92.6_{\pm 0.2}$ | $88.9_{\pm 0.4}$ | 90.8 |
| CCL + Few-Shot | $93.5_{\pm 0.3}$ | $89.7_{\pm 0.3}$ | $\mathbf{93.3}_{\pm 0.1}$ | $90.8_{\pm 0.1}$ | 91.8 |
| OE + Wikitext | $92.6_{\pm 0.8}$ | $88.9_{\pm 0.4}$ | $91.6_{\pm 0.6}$ | $89.8_{\pm 0.1}$ | 90.7 |
| OE + Novelty Prompting | $83.6_{\pm 0.4}$ | $\mathbf{90.6}_{\pm 0.2}$ | $92.4_{\pm 0.1}$ | $\mathbf{91.3}_{\pm 0.3}$ | 89.5 |
| Contrastive Novelty Learning | $\mathbf{94.3}_{\pm 0.2}$ | $90.5_{\pm 0.3}$ | $\mathbf{93.4}_{\pm 0.1}$ | $\mathbf{91.1}_{\pm 0.2}$ | $\mathbf{92.3}$ |
| CCL + Gold Label † | $94.8_{\pm 0.3}$ | $91.4_{\pm 0.3}$ | $93.7_{\pm 0.1}$ | $91.0_{\pm 0.2}$ | 92.7 |
| CCL + Gold Data † | $96.6_{\pm 0.1}$ | $93.5_{\pm 0.1}$ | $94.8_{\pm 0.2}$ | $94.3_{\pm 0.4}$ | 94.8 |
| OE + Gold Data † | $96.5_{\pm 0.2}$ | $94.8_{\pm 0.0}$ | $95.2_{\pm 0.0}$ | $96.2_{\pm 0.2}$ | 95.7 |

Table 1: *OSSC Results of Contrastive Novelty Learning.* Methods listed below CNL are upper bounds. All outlier exposure (OE) methods are trained on 100K outlier generations. We average over the results of 5 seeds of all splits and report standard error of the mean in subscript. We report macro-average of all datasets in the rightmost column. Oracle methods are marked with a †. We find that CNL significantly outperforms all prior methods on 3 of 4 datasets, and both the Novelty Prompting and CCL loss components are important for strong performance.

## 4.3 BASELINES

We evaluate our method against OSSC baselines from prior work, CCL baselines with other novel sets, and Outlier Exposure (Hendrycks et al., 2019). For all methods, we train a BERT-base model and use hyperparameters from the original papers unless otherwise specified. Of the basleines, only CCL and Outlier Exposure use an explicit novel set.

**Vanilla** We evaluate vanilla cross-entropy loss training, calculating confidence using MaxProb.

**kFolden** We evaluate kFolden (Li et al., 2021), a method that trains an ensemble of $k$ individual classifiers, each trained on $k - 1$ labels. The average of the ensemble probability distributions is used for confidence computation.

**Contrastive** We evaluate Contrastive OOD Detection (Zhou et al., 2021), which uses a contrastive objective to induce training examples of different classes to be distant and of the same class to be near. This sparsifies the embedding space, ensuring that most OOD examples are far from feature representations of ID samples. We use the supervised constrastive loss and the Mahalanobis distance metric for confidence computation, finding that this setup performed the best on our evaluation.

**CCL + Zero/Few-Shot Data Augmentation** To measure the impact of explicitly prompting for novel labels, we generate with an identical pretrained GPT-J model, but prompt with only an instruction and one (or zero) ID training example from each class (See Appendix A.1 for the specific prompt format). Essentially, we perform example generation identically but skip label generation entirely. We perform CCL training and MaxProb inference. While some resultant generations will be useful, we expect that many will not be semantically novel, resulting in strictly worse performance.

**CCL + Wikitext** To measure whether plausibility of examples impacts their usefulness for CCL, we use an entirely different dataset, Wikitext-103, as our novel set. Though these examples represent a distribution shift, they do not accurately reflect the open-set shift the classifier will encounter.

**Outlier Exposure + Novelty Prompting** We pair our novel set with Outlier Exposure (OE) (Hendrycks et al., 2019) as described in Section 3.2 and compute confidence with MaxProb.

## 5 RESULTS

### 5.1 OSSC RESULTS

**CNL outperforms prior work.** We report comparisons of CNL against baselines in Table 1. Broadly, we find that while baselines like kFolden and Contrastive training struggle to consistently

| Label | Generated Example |
|---|---|
| CURIOSITY | i am still interested but more interested to visit the pyramids and learn more |
| DESPAIR | i love my friends but sometimes i feel like im not good enough |
| DISAPPOINTMENT | i am a human nothing is going to keep me from flying away |

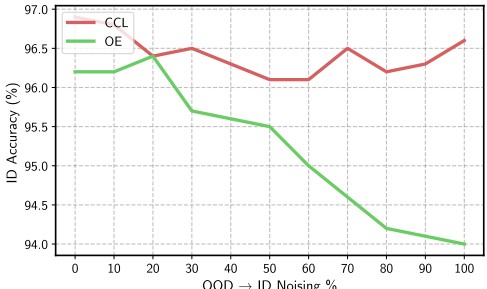

Figure 2: *Example novel generations for Emotion*. In this split, the gold novel label is "sadness". Though we remove the gold novel label before example generation, many generations are still relevant to this label. More generation examples are shown in Appendix A.4.

Figure 3: *Noisy novel sets hurt accuracy for OE*. We plot the ID accuracy of classifiers trained with OE and CCL on mixtures of held-out TREC-10 OOD and ID data as an novel set. ID accuracy with OE decreases as we introduce more noise, while CCL stays stable.

outperform vanilla training (e.g., on TACRED), CNL improves selective classification over vanilla across all datasets. We outperform the best prior method (Contrastive) by 2.3% AUAC, and on three of four datasets, our method significantly outperforms *all* prior methods. Furthermore, we outperform kFolden by 3.6% AUAC despite its ensemble totaling many times the capacity of our single classifier. CNL also results in zero or insignificant accuracy drop (less than 0.2 points) for all datasets. In Appendix A.2, we show full ID accuracy results for all datasets.

**Other choices of novel set for CCL training can still be beneficial.** Prompting with only a task-relevant instruction (zero-shot) generates sufficiently useful novel examples to slightly outperform the vanilla baseline by 1.5% AUAC. Using Wikitext as our novel set performs roughly on par with zero-shot generation: though Wikitext examples are less noisy than generations, they also tend to be less dataset-relevant. Few-shot generation, which generates more plausible examples, is significantly better than all prior methods, though it performs worse than Novelty Prompting on 3 of 4 datasets.

To further test the importance of novel set selection, we compare with two oracle methods. In the Gold Data setting, we train with CCL against held out data of the gold novel test class(es) as a strict upper bound for both label and example generation. In the Gold Label setting, we eliminate the label generation step, fixing the gold label as the sole member of our novel label set. We perform example generation as normal using this gold label alone. This setting is overly optimistic as we cannot know what new labels will appear at test-time.[4] We find that CCL in the Gold Label setting slightly outperforms CNL, but paired with gold novel data can achieve significantly stronger OSSC.

**Training loss choice matters for generated data.** Although OE training with Novelty Prompting data improves OOD detection over vanilla, it causes a sharp decrease in accuracy on TREC-10 (96.6% → 71.3%) and an average of 0.6% accuracy on the other three datasets (see Appendix A.2). We attribute this behavior to generation noise: some generated examples are similar to ID examples, and thus greatly affect the model's ID predictions when training with OE. To verify this hypothesis, we conduct an experiment where we train classifiers with novel sets formed by noising heldout OOD data with various amounts of heldout ID data. In Figure 3, we show that as the simulated ID noise ratio increases, OE training hurts accuracy, whereas CCL models retain accuracy. Though this phenomenon is most salient on TREC-10, we show in Appendix A.10 that it also occurs on other small datasets, and is more severe when the novel set is larger than the training set.

When the novel set is sampled from held-out gold OOD data instead of being generated by a LLM, OE outperforms CCL in AUAC and AUROC, suffering only a small accuracy drop (an average of 0.4%). In contrast, we find that CCL training maintains accuracy on all generated novel data settings (see Appendix A.2), as it does not enforce a uniform probability distribution on all novel set examples. CCL with both zero- and few-shot augmented novel sets outperforms all prior methods, and our full CNL method (CCL+NP) significantly outperforms prior methods on all but one dataset.

---

[4]In practice, expert knowledge of the novelty we expect to see at test-time is sometimes available, and as shown in our results, can be leveraged for better performance.

| AUROC (↑) | TREC-10 | AGNews | Emotion | TACRED | Average |
|---|---|---|---|---|---|
| Vanilla | $76.6_{\pm 4.4}$ | $76.4_{\pm 1.0}$ | $85.0_{\pm 2.4}$ | $46.3_{\pm 0.1}$ | 71.1 |
| kFolden | $84.7_{\pm 2.0}$ | $72.5_{\pm 2.2}$ | $85.3_{\pm 1.8}$ | $\mathbf{53.1}_{\pm 6.2}$ | 73.9 |
| Contrastive | $79.8_{\pm 1.3}$ | $76.5_{\pm 1.8}$ | $89.1_{\pm 1.7}$ | $45.7_{\pm 1.2}$ | 72.3 |
| CCL + Wikitext | $81.0_{\pm 2.6}$ | $78.1_{\pm 0.8}$ | $90.3_{\pm 0.8}$ | $45.2_{\pm 1.2}$ | 74.1 |
| CCL + Zero-Shot | $84.8_{\pm 1.4}$ | $78.8_{\pm 0.8}$ | $90.7_{\pm 0.7}$ | $44.2_{\pm 1.0}$ | 74.6 |
| CCL + Few-Shot | $88.4_{\pm 0.6}$ | $80.5_{\pm 0.7}$ | $\mathbf{92.8}_{\pm 0.5}$ | $49.7_{\pm 0.3}$ | 77.9 |
| OE + Wikitext | $85.0_{\pm 1.7}$ | $78.3_{\pm 0.8}$ | $88.8_{\pm 1.1}$ | $46.2_{\pm 0.5}$ | 74.6 |
| OE + Novelty Prompting | $74.2_{\pm 0.5}$ | $\mathbf{85.5}_{\pm 0.3}$ | $91.0_{\pm 0.3}$ | $\mathbf{53.5}_{\pm 0.7}$ | 76.0 |
| Contrastive Novelty Learning | $\mathbf{90.8}_{\pm 0.6}$ | $82.6_{\pm 0.6}$ | $\mathbf{93.4}_{\pm 0.3}$ | $50.9_{\pm 0.5}$ | 79.4 |
| CCL + Gold Label † | $92.0_{\pm 0.8}$ | $84.9_{\pm 0.4}$ | $94.2_{\pm 0.3}$ | $51.2_{\pm 0.6}$ | 80.6 |
| CCL + Gold Data † | $98.3_{\pm 0.3}$ | $91.7_{\pm 0.3}$ | $98.8_{\pm 0.1}$ | $63.1_{\pm 0.2}$ | 88.0 |
| OE + Gold Data † | $99.1_{\pm 0.2}$ | $98.8_{\pm 0.3}$ | $99.7_{\pm 0.0}$ | $89.0_{\pm 0.5}$ | 96.7 |

Table 2: *OOD Detection Results of Contrastive Novelty Learning.* Methods same as in Table 1. We find that CNL signficantly improves OOD detection AUROC over all prior methods on 3 of 4 datasets. While OE training results in better AUROC on some datasets, it hurts ID accuracy.

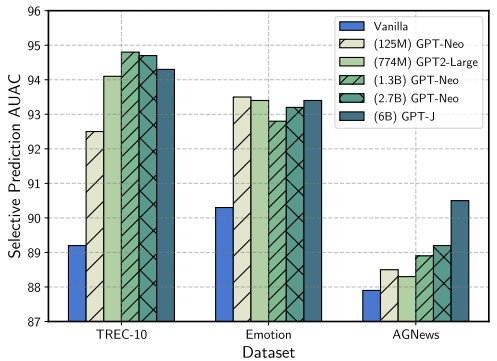

Figure 4: *Smaller generators can also help.* We perform generation with variously sized generator models, from 125M to 6B parameters. Larger generators seem to yield better results, but even our smallest generator does well.

Figure 5: *Larger classifiers can also benefit.* We additionally train RoBERTa-base and RoBERTa-large with and without Novelty Prompted training. We find that both classifiers improve over vanilla with CNL.

## 5.2 OOD DETECTION RESULTS

To confirm that CNL improves a classifier's ability to disambiguate novel class examples, we compare CNL against the same baselines on OOD detection in Table 2. We find similar improvements, outperforming the best prior method (kFolden) by 5.5% AUROC. We interpret this result in Appendix A.11, showing that CNL improves ID/OOD separability. Unlike other datasets, TACRED exhibits strong OOD overconfidence: all baselines except kFolden yield *worse*-than-random OOD detection (below 50% AUROC). We hypothesize that this could be due to models incorrectly assuming that an NER tag pair seen at training time in only a single class could not belong to a novel relation. OOD detection on TACRED remains a challenging goal for future work, as the strong performance of CCL training with gold heldout data indicates significant remaining headroom. In fact, on all three other datasets, models achieve greater than 90% AUROC when trained with gold heldout data. While OE results in better AUROC performance on AGNews, ID accuracy also decreases.

## 5.3 PERFORMANCE ANALYSIS

**Label Generator Model** We investigate whether a smaller, open-source model can suffice as the label generator. Specifically, we replace the label generator with GPT-J and use 100 label generation iterations. We find that GPT-J performs on-par with GPT-3 on 3 of 4 datasets in all metrics, except on AGNews where it performs within 1 point AUAC. We provide full details in Appendix A.3.

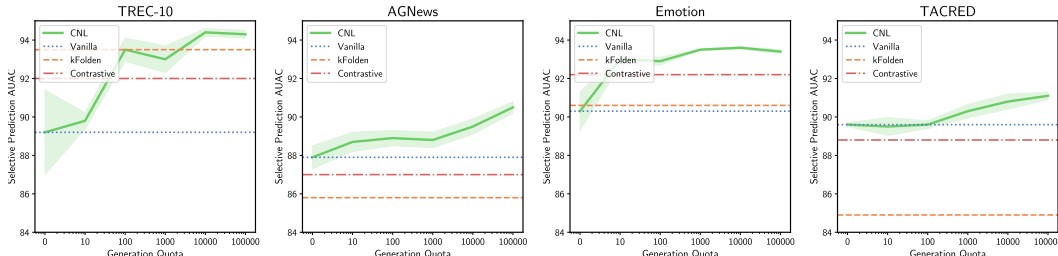

Figure 6: *Selective prediction performance is positively correlated with generation quota.* We measure selective prediction against the total number of examples generated, showing various baselines in horizontal lines. On most datasets, even a low quota can meaningfully improve AUAC.

**Example Generator Size** Since model scale often affects prompting performance Sanh et al. (2021); Brown et al. (2020), we compare generator models ranging in size from 125M parameters to 6B parameters. For each, we generate 100K examples, and compare CNL results in Figure 4. All generators improve over the Vanilla baseline. GPT2-Neo 125M is competitive with GPT2-Large despite being roughly 5x smaller, suggesting that its larger pretraining corpus (the Pile) aids generation ability. We observe that novel generation is easier on simpler tasks: on Emotion, where labels (or synonyms) can appear directly in the example, inter-generator differences are small. We posit that even larger generators such as GPT-3 could yield better performance across abstract tasks.[5]

**Other Classifier Models** We investigate the generalizability of CNL to two other classifier architectures, RoBERTa (Liu et al., 2019) and DeBERTa-v3 (He et al., 2021), of both base and large sizes, with results in Figure 5. Averaged over datasets, CNL improves AUAC for all classifiers, though these improvements are most apparent with the smaller base models. Larger classifiers are better at OSSC: vanilla RoBERTa-large improves over BERT-base by 2.8% AUAC. Vanilla RoBERTa-base slightly outperforms vanilla BERT-base, but after CNL training, the two perform on-par, suggesting that learning from generated examples can make up for BERT's smaller pretraining corpus.

**Generation Quota** Since large-scale LLM prompting is costly, we analyze the performance tradeoff of shrinking the generation quota, the number of novel examples that we can generate. In Figure 6, we show that on some datasets, using orders of magnitude smaller novel sets can still improve selective prediction. For example, 1000 generations is sufficient to improve AUAC across all datasets, and for most datasets we require far fewer. In such cases where a low quota is sufficient, novelty prompting is nearly as efficient as vanilla classifier training.

**Generation Analysis** To evaluate the remaining errors in OOD generation, we perform two types of manual analysis on Novelty Prompting (NP). First, we categorize the labels generated by NP after filtering, finding that 70%+ of GPT-3 generated labels are novel on all datasets except TREC-10, where only 40% are novel, and the vast majority of the others are valid closed-set examples. This highlights one source of generation noise in our pipeline. Second, we categorize the examples generated by NP and a well-performing baseline method, Few-shot data augmentation (FS). Specifically, for each of 4 splits of AGNews, we annotate 100 NP and 100 FS examples. We find that on average 41% of NP generations come from novel classes, compared to only 26% of FS generations, explaining CNL's stronger performance over CCL + Few-Shot. We provide further analysis in Appendix A.5. Our method performs well despite the high fraction (50.5%) of closed-set examples generated in NP, showing that CCL is robust to noise in the novel example generation process.

## 6 RELATED WORK

### 6.1 IDENTIFYING OOD DATA

**OOD Detection.** Significant prior work has focused on OOD detection, which asks a model to detect when a test example originates from a new distribution (Hendrycks & Gimpel, 2017). Many of these change the training objective of a model, e.g., with a contrastive training objective (Winkens et al.,

---

[5]We evaluate GPT-3 for label generation but not for example generation, as the latter requires many more API calls.

2020; Sehwag et al., 2021; Zhou et al., 2021). When the nature of the distribution shift is known, the model can directly be trained to be uncertain on known OOD examples (Dhamija et al., 2018; Hendrycks et al., 2019). We draw on the success of these known-shift methods, but eliminate the need for known OOD data by using generative models.

Recently, many works on OOD detection have explored alternative modeling paradigms. By drawing on assumptions about the IID nature of neural network errors, confidence can be predicted from model ensembles (Ţifrea et al., 2021; Li et al., 2021; Lakshminarayanan et al., 2017). Simple methods like deep nearest-neighbors Sun et al. (2022); Bergman et al. (2020) can also perform surprisingly well. Further performance improvements can be achieved by modifying the confidence metric. Podolskiy et al. (2021) find that Mahalanobis distance better exploits the geometry of the learned embedding space, explaining strong performance achieved by replacing probability-based scoring mechanisms (Lee et al., 2018; Ren et al., 2021). We show that standard models are sufficient: Max-Prob scoring with a standard classifier can perform well when given proper OOD demonstrations.

**OOD Selective Prediction.** Selective prediction work focuses on a different paradigm altogether, fusing abstention (detection) with prediction (El-Yaniv & Wiener, 2010; Geifman & El-Yaniv, 2017). External calibrators popularized by Kamath et al. (2020) have become popular as a selective prediction framework Zhang et al. (2021); Ye & Durrett (2021); Varshney et al. (2022). However, calibrators are typically smaller than classifier models (Tajwar et al., 2021), limiting their capacity to learn the volume of outliers we generate and swaying us to consider using no auxiliary model to detect novel examples. Additionally, methods such as ours which train a model to predict their own confidence can be deployed with no additional inference overhead. For example, large generative LMs can be trained to self-verify on challenging question answering tasks (Kadavath et al., 2022).

## 6.2 OPEN-SET CLASSIFICATION

Open-set classification is well-explored in the image classification space, as tasks like CIFAR-100 tend towards large label spaces (Scheirer et al., 2013; Geng et al., 2021). Some methods for detecting open-set examples build on the classifier, e.g., by classifying over the model's activations (Bendale & Boult, 2016) or adding an additional reconstruction model (Oza & Patel, 2019). Our work is most closely related to methods which generate near-OOD examples and regularize confidence on them (Ge et al., 2017; Du et al., 2022; Kong et al., 2020; Vernekar et al., 2019; Möller et al., 2021; Setlur et al., 2022). However, methods like perturbation and embedding space sampling align poorly with the discrete nature of text, prompting us to investigate powerful generative language models. Esmaeilpour et al. (2022) is closely related to our work in that they also generate novel labels, but directly use these labels as input to a classifier.

Open-set classification for text has been less explored. Early works built upon the $k$-way, 1-vs-rest paradigm of SVMs, classifying an example as "novel" if all $k$ scores fall below a threshold (Fei & Liu, 2016; Shu et al., 2017; Doan & Kalita, 2017). Many recent works explore similar methods as prior vision work, but focus on the intent detection setting, as task-oriented dialogue models should abstain on unknown intents (Zeng et al., 2021; Zheng et al., 2020; Lin & Xu, 2019). To the best of our knowledge, we are the first work to generate novel examples for open-set text classification.

## 7 DISCUSSION AND FUTURE WORK

In this work, we introduce Contrastive Novelty Learning, a method for generating novel examples which simulate open-set shift and training to abstain on them. Through extensive experiments, we demonstrate that by presenting generated examples to a classifier, we can significantly improve its ability to abstain on examples from novel classes against state-of-the-art baselines. Our work provides a generalizable framework for improving OSSC and OOD detection: in fact, we show through CCL training's strong performance with gold heldout data that there remains significant headroom for novel example generation. Additionally, CNL is modular, as it provides additional supervision signal but does not alter the classifier model's architecture, and thus remains extensible with other training objectives or classification metrics. Finally, given recent interest in the emergent capabilities of large language models, we hope that future work on classification in the presence of distribution shifts can better leverage large language models to both directly identify shifts and improve the abstention ability of smaller classifiers.

## 8 REPRODUCIBILITY STATEMENT

We provide a full code implementation of Contrastive Novelty Learning, including Novelty Prompting and CCL and OE training. We detail hyperparameters for our experiments in Section 4.2, prompt formats in Appendix A.1, and data processing details in Appendices A.6 and A.7. CNL can be run with any text dataset and discriminative classifier model. We release generated novel examples from Novelty Prompting and several generation baselines for reproducibility.

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

# A  APPENDIX

## A.1  PROMPT FORMAT

We use the same format for label generation for all datasets, shown in Figure 7, but customize the instruction for each dataset, as shown in Figure 8.

| | |
|---|---|
| Instruction | `Generate a diverse list of news genres:` |
| ID Labels | `[World, Sports, Sci/Tech,` |

Figure 7: *Label Generation prompt for AGNews.*

| Dataset | Instruction |
|---|---|
| Emotion | `Generate a diverse list of emotions` |
| AGNews | `Generate a diverse list of news genres` |
| TREC-10 | `Generate a diverse list of entity types` |
| TACRED | `Generate a diverse list of relations between entities` |

Figure 8: *Label Prompts for each Dataset.*

For example generation, we prompt with an example sampled from each class and a random novel label. We use the same instruction for all datasets. An example prompt is shown in Figure 9.

| | |
|---|---|
| Instruction | `Given a label, generate a corresponding example:` |
| ID Label 1 | `business` |
| ID Example 1 | `Starwood Names New Chief Executive SEPTEMBER 21,` |
| | `2004 -- White Plains, NY -- Former Coca-Cola Company` |
| | `president Steven Heyer today was named the new chief` |
| | `executive of Starwood Hotels, effective Oct. 1.` |
| | `Heyer succeeds Starwood founder Barry` |
| ID Label 2 | `sports` |
| ID Example 2 | `Marino, Young Considered for Hall of Fame Dan Marino` |
| | `and Steve Young highlighted a list Friday of 25` |
| | `candidates for the Pro Football Hall of Fame.` |
| ID Label 3 | `world` |
| ID Example 3 | `Afghan warlords 'threaten poll' Afghan warlords` |
| | `are involved in intimidation which could threaten` |
| | `October's elections, Human Rights Watch says.` |
| Novel Label | `entertainment` |

Figure 9: *Example Generation prompt for AGNews.*

Few-shot prompting is done with a task-specific instruction, but does not include labels, as shown in Figure 10. Zero-shot prompting is done with the task-specific instruction only.

## A.2  FULL ACCURACY RESULTS

CNL improves AUAC on all datasets without any cost to ID accuracy, as shown in Figure 11. We show full ID accuracy results in Table 3. CCL training maintains accuracy across all datasets,

| Instruction | Generate a news headline: |
|---|---|
| ID Example 1 | Starwood Names New Chief Executive SEPTEMBER 21, 2004 -- White Plains, NY -- Former Coca-Cola Company president Steven Heyer today was named the new chief executive of Starwood Hotels, effective Oct.  1.  Heyer succeeds Starwood founder Barry |
| ID Example 2 | Marino, Young Considered for Hall of Fame Dan Marino and Steve Young highlighted a list Friday of 25 candidates for the Pro Football Hall of Fame. |
| ID Example 3 | Afghan warlords 'threaten poll' Afghan warlords are involved in intimidation which could threaten October's elections, Human Rights Watch says. |

Figure 10: *Few-Shot Generation prompt for AGNews.*

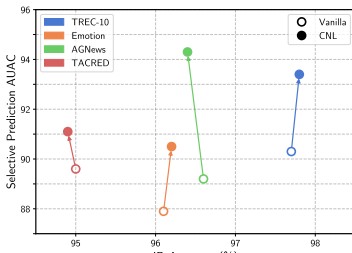

|  | (↑) | TREC-10 | AGNews | Emotion | TACRED | Avg |
|---|---|---|---|---|---|---|
| Vanilla | AUAC | $89.2_{\pm2.2}$ | $87.9_{\pm0.6}$ | $90.3_{\pm1.0}$ | $89.6_{\pm0.1}$ | 89.3 |
|  | AUROC | $76.6_{\pm4.4}$ | $76.4_{\pm1.0}$ | $85.0_{\pm2.4}$ | $46.3_{\pm0.1}$ | 71.1 |
|  | ID Acc | $96.6_{\pm0.2}$ | $96.1_{\pm0.0}$ | $97.7_{\pm0.1}$ | $95.0_{\pm0.1}$ | 96.4 |
| GPT-3 | AUAC | $94.3_{\pm0.2}$ | $90.5_{\pm0.3}$ | $93.4_{\pm0.1}$ | $91.1_{\pm0.2}$ | 92.3 |
|  | AUROC | $90.8_{\pm0.6}$ | $82.6_{\pm0.6}$ | $93.4_{\pm0.3}$ | $50.9_{\pm0.5}$ | 79.4 |
|  | ID Acc | $96.4_{\pm0.2}$ | $96.2_{\pm0.0}$ | $97.8_{\pm0.1}$ | $94.9_{\pm0.1}$ | 96.3 |
| GPT-J | AUAC | $94.2_{\pm0.3}$ | $89.8_{\pm0.3}$ | $93.5_{\pm0.1}$ | $91.0_{\pm0.2}$ | 92.1 |
|  | AUROC | $90.0_{\pm0.6}$ | $80.8_{\pm0.6}$ | $93.5_{\pm0.3}$ | $50.4_{\pm0.4}$ | 78.7 |
|  | ID Acc | $96.4_{\pm0.1}$ | $96.2_{\pm0.0}$ | $97.9_{\pm0.1}$ | $94.9_{\pm0.0}$ | 96.4 |

Figure 11: *CNL training maintains accuracy.* Training with novelty prompted examples does not significantly alter ID accuracy, but improves selective prediction across all datasets.

Figure 12: *GPT-J is also a strong label generator.* We compare label generation using GPT-3 and GPT-J, using GPT-J as the example generator for both methods. GPT-J performs within a negligible margin of GPT-3 on TREC-10 and Emotion, but slightly worse on AGNews and TACRED.

while OE training decreases accuracy on 3 of 4 datasets, with a very sharp drop on TREC-10. In Appendix A.10, we show through analyses on two datasets that this steep accuracy drop is not an anomaly: when paired with generated data, OE training is sensitive to the sizes of the novel set and training set, and can signficantly hurt ID accuracy when the novel set is much larger than the training set. Additionally, despite improving selective prediction performance, training with gold held-out data curiously hurts accuracy on TACRED.

### A.3   CNL PERFORMS WELL WITHOUT GPT-3

In our main experiments, we use GPT-3 as the label generator and GPT-J 6B as the example generator. In Section 5.3, we show that smaller models can be used as *example generators*. Here we investigate whether a smaller, open-source language model can be used as a *label generator*. In Table 12, we show that GPT-J 6B also performs well at label generation. We empirically observe that GPT-J generates shorter and noisier completions, requiring us to increase the number of model calls from 5 to 100 and filter out all labels containing punctuation marks. We find that the difference between GPT-J and GPT-3 label generation is only significant on 1 of 4 datasets (AGNews), suggesting that CNL with GPT-J only can still work well.

### A.4   GENERATION EXAMPLES

We show examples of the generations from Novelty Prompting for AGNews in Table 4. Recall that we do not allow the gold novel label to be generated to hedge against data leakage from LLM pretraining. However, we observe that our generator is still capable of producing relevant examples to the gold novel label due to signal from similar novel labels. Despite many generations not being directly relevant to the gold novel label, we observe that the generated novel labels are sufficiently

| ID Acc (↑) | TREC-10 | AGNews | Emotion | TACRED |
|---|---|---|---|---|
| Vanilla | $96.6_{\pm 0.2}$ | $96.1_{\pm 0.0}$ | $97.7_{\pm 0.1}$ | $95.0_{\pm 0.1}$ |
| kFolden | $96.5_{\pm 0.1}$ | $96.0_{\pm 0.1}$ | $97.2_{\pm 0.2}$ | $88.3_{\pm 0.0}$ |
| Contrastive | $95.3_{\pm 0.1}$ | $96.0_{\pm 0.0}$ | $98.0_{\pm 0.1}$ | $94.8_{\pm 0.2}$ |
| CCL + Wikitext | $96.6_{\pm 0.1}$ | $96.1_{\pm 0.1}$ | $97.6_{\pm 0.1}$ | $94.9_{\pm 0.1}$ |
| CCL + Zero-Shot | $96.5_{\pm 0.2}$ | $96.3_{\pm 0.0}$ | $97.6_{\pm 0.1}$ | $94.9_{\pm 0.2}$ |
| CCL + Few-Shot | $96.3_{\pm 0.2}$ | $96.1_{\pm 0.0}$ | $97.8_{\pm 0.1}$ | $94.8_{\pm 0.2}$ |
| OE + Wikitext | $96.6_{\pm 0.2}$ | $96.1_{\pm 0.0}$ | $97.6_{\pm 0.1}$ | $94.8_{\pm 0.1}$ |
| OE + Novelty Prompting | $71.3_{\pm 0.9}$ | $95.6_{\pm 0.0}$ | $96.4_{\pm 0.2}$ | $94.8_{\pm 0.2}$ |
| Contrastive Novelty Learning | $96.4_{\pm 0.2}$ | $96.2_{\pm 0.0}$ | $97.8_{\pm 0.1}$ | $94.9_{\pm 0.1}$ |
| CCL + Gold Label † | $96.5_{\pm 0.2}$ | $96.1_{\pm 0.0}$ | $97.7_{\pm 0.1}$ | $94.9_{\pm 0.1}$ |
| CCL + Gold Data † | $96.0_{\pm 0.2}$ | $95.8_{\pm 0.1}$ | $97.6_{\pm 0.1}$ | $93.8_{\pm 0.1}$ |
| OE + Gold Data † | $96.4_{\pm 0.1}$ | $95.8_{\pm 0.0}$ | $97.9_{\pm 0.1}$ | $93.6_{\pm 0.2}$ |

Table 3: *Full Accuracy Results of Contrastive Novelty Learning*

| Dataset | Label Type | Frequency | Example Label |
|---|---|---|---|
| TREC-10 | Implausible | 7.8% | August 27 |
|  | Novel | 40.0% | time |
|  | Closed-Set | 52.2% | person |
| AGNEWS | Implausible | 14.9% | ology |
|  | Novel | 81.7% | food |
|  | Closed-Set | 3.3% | technology |
| EMOTION | Implausible | 5.1% | app |
|  | Novel | 83.9% | serenity |
|  | Closed-Set | 11.1% | frustration |
| TACRED | Implausible | 14.3% | ualifications |
|  | Novel | 73.6% | parent company |
|  | Closed-Set | 12.1% | current location |

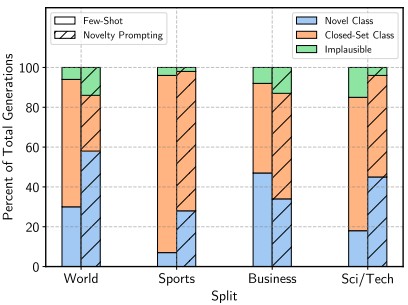

Figure 13: *Error Analysis on Label Generation.* We manually annotate generated label sets across all splits of each dataset, recording the frequency of novel and plausible labels.

Figure 14: *Error Analysis on Example Generation.* For each split of AGNews, we manually annotate 100 generations each for two generation methods and compute the frequency of novel class examples, closed-set class examples, and implausible examples.

distinct from the closed-set labels that most generated examples still provide useful "novelty" supervision signal to the classifier.

## A.5 NOVELTY PROMPTING ERROR ANALYSIS

Though CNL improves OSSC ability on all datasets, we still find headroom between Novelty Prompting generated data and gold OOD data ($92.3 \rightarrow 94.8$) in Table 1. To understand the remaining failure modes of Novelty Prompting, we manually inspect the generated labels and examples from our method. Broadly, we seek to attribute "generation noise," or the frequency with which the purported novel sets which we generate instead contain closed-set class examples.

First, we manually annotate GPT-3 generated labels from all dataset splits, categorizing a label into "implausible" if it does conform to the dataset's format, "closed-set" (ID) if it is synonymous with a class seen in training, and "novel" (OOD) if it describes a class distinct from all closed-set classes. In Figure 13, we perform this analysis for all four datasets. Across all datasets, less than 15% of generations are implausible, suggesting that the model is usually able to generate reasonable additional labels given only 3-6 ID classes. We also observe that while on 3 of 4 datasets less than 15% of generated classes are closed-set, on TREC-10 more than half of generated labels are closed-set. One reason for this label generation noise is that the TREC-10 labels are very broad (e.g., "entity" describes questions about any subcategory of an entity, including all objects and events), so

| Gold Label | Generated Label | Generated Example |
|---|---|---|
| WORLD | HEALTH | The Top 5 Places to Take a Break From the Rat Race This summer could be the best one in years to take a break and recharge the ol' batteries. |
| | TRAVEL | U.S. Airlines Say They Won't Raise Rates to Pay for the New Security Fee and The government, for its part, has set a deadline for some of the other fees at the beginning of this month. |
| | POLITICS | The Real Reason Republicans Cautiously Endorse the 'Obscure' Paul Ryan - CNN Politics If we're going to pick fights with the other party, we might as well pick fights with the right people. |
| SPORTS | ENTERTAINMENT | David Beckham And Victoria Beckham Are Exiting The London Home They Have Shared With The NBA Star. LONDON (AP) - David Beckham has been kicked out of the house that he shared with his British soccer team, the Los Angeles Galaxy. The couple and the couple's kids are moving into an apartment |
| | FASHION | In search of fashion in South Africa What will the fashion world be wearing next year? Will the world need another white wedding dress? |
| | CRIME | The Police Are Going To Find You a Lot Easier in the Future: Security Advances of 2008 (Video) The police are going to find you a lot easier in the future: Security advances of 2008 |
| BUSINESS | FOOD/DRINK | How to Make a Fried Sausage Slider (Fast) - YouTube A couple of weeks ago, I made a fried sausage slider and they're awesome! A few friends have asked me to go through this again but in a shorter time to make up a batch quickly for them. |
| | WEATHER | Snowstorm Blasts Northeast with Another Round of Winter Storms to Leave Snow, Ice and Blowdowns A big nor'easter is expected to bring sleet, ice and another round of dangerous wind gusts around New Year's Eve. Winter storms are expected to roll through Wednesday and Thursday in the |
| | RELATIONSHIPS | AP - Michael Phelps and his wife will be moving back to Washington State from Arizona while he finishes his Olympic career. |
| SCI/TECH | LIFESTYLE | A new batch of Apple iPhone 3Gs have gone up for sale in the UK, with all six major networks now having a network price. Apple unveiled the 3Gs on Wednesday, making a number of changes to the device, which is expected to be hugely popular in the market. |
| | ENTERTAINMENT | THE FILM: JERSEY GIRL "Jersey Girl" tells the story of the love-hate relationship between an Irish-American girl from New Jersey and a native New Jerseyan. Directed by Elizabeth Swados. |
| | TECHNOLOGY | Yahoo Japan to buy a majority stake in Nikkei Corp Yahoo Japan Corporation announced it plans to buy a 69.8 per cent stake of Nikkei for 1.43billion, the two companies said Friday. |

Table 4: *Example novel generations for AGNews.*

while a generated label might differ in definition, it could still overlap with or fall into a subcategory of a closed-set class.

Second, we manually annotate GPT-J generated examples to understand whether example generation is a source of generation noise. In Figure 14, we annotate 100 examples of each split of AGNews for both Few-shot data augmentation and Novelty Prompting. We observe that Novelty Prompting generates novel class examples significantly more frequently across 3 of 4 splits. Both methods generate implausible (e.g., agrammatical, non-news) examples rarely, as ID demonstrations sufficiently prime the model to generate text in the style of news. Additionally, under Novelty Prompting, we find that the fraction of novel class examples (41.3%) is significantly lower than the fraction of novel labels generated (81.7%), suggesting that GPT-J can easily adhere to the dataset format, but struggles to extrapolate to the novel label. Future work should thus focus on better specifying the example generation step to leverage the generated labels.

| | (↑) | TREC-10 | AGNews | Emotion | TACRED | Avg |
|---|---|---|---|---|---|---|
| **Vanilla** | AUAC | $89.2_{\pm2.2}$ | $87.9_{\pm0.6}$ | $90.3_{\pm1.0}$ | $89.6_{\pm0.1}$ | 89.3 |
| | AUROC | $76.6_{\pm4.4}$ | $76.4_{\pm1.0}$ | $85.0_{\pm2.4}$ | $46.3_{\pm0.1}$ | 71.1 |
| | ID Acc | $96.6_{\pm0.2}$ | $96.1_{\pm0.0}$ | $97.7_{\pm0.1}$ | $95.0_{\pm0.1}$ | 96.4 |
| **LS** | AUAC | $90.6_{\pm1.6}$ | $83.5_{\pm1.4}$ | $82.0_{\pm1.7}$ | $87.1_{\pm1.0}$ | 85.8 |
| | AUROC | $80.5_{\pm3.7}$ | $72.9_{\pm1.7}$ | $75.1_{\pm2.3}$ | $41.2_{\pm2.4}$ | 67.4 |
| | ID Acc | $96.7_{\pm0.2}$ | $96.2_{\pm0.0}$ | $97.7_{\pm0.1}$ | $95.0_{\pm0.1}$ | 96.4 |

Table 5: *Label Smoothing (LS) hurts AUROC and AUAC on all but one dataset.*

## A.6 DATASET SPLIT DETAILS

**TREC-10**: We remove the `Abbreviation` class as it is too small to yield statistically significant metrics in our task setting, leaving 5 remaining classes.

**Emotion** (Saravia et al., 2018): We remove two small classes, `love` and `surprise`, leaving 4 remaining classes.

**TACRED** (Zhang et al., 2017): We process the data for training following Joshi et al. (2019). This dataset is particularly challenging due to its class-imbalanced nature. We evaluate a single split where we keep the 6 largest classes as ID data, and hold out the other 35. This is the largest class, and thus results in approximately $80\%$ of examples being OOD at test time.

## A.7 TACRED PROCESSING DETAILS

We perform label normalization, removing underscores and prefixes, e.g., converting `per:employee_of` into `employee of`. This both helps the label generator model understand our label space and generate more relevant novel labels and ensures that generated novel labels are well-formatted for downstream example generation.

For examples, we normalize the Subject and Object token tags into a standard English equivalent containing the subject or object indicator and the NER tag, e.g., [`subject:person`]. To ensure that generated examples satisfy the task format, we filter out examples that do not contain exactly one subject and one object (many generations contain partial or malformed indicator/NER spans). Finally, we denormalize tags back into original model input tokens.

## A.8 LABEL FILTERING

After label generation, we perform synonym filtering to reduce occurrences of ID synonyms. We find this step to have a large impact on datasets for which labels are common English words which appear in our thesaurus, and less where label names are more abstract. For example, for Emotion and TREC-10 , where dataset names are words such as "fear" or "human," filtering removes 21% and 20% of generated labels respectively. Meanwhile on both AGNews and TACRED, label filtering removes only 2% of labels. In the case of AGNews, news genre overlaps are not easily captured by synonyms, and even after normalization, many TACRED labels such as "employee of" do not appear in our thesaurus.

## A.9 LABEL SMOOTHING PERFORMS POORLY.

We evaluate label smoothing (LS) (Müller et al., 2019) as an additional baseline for improving OSSC, which mirrors vanilla training but alters the one-hot target vector to a "smoother" version, incentivizing uncertainty. Label smoothing has been shown to be effective in domain shift detection (Kong et al., 2020). We use label smoothing factor $\alpha = 0.1$ and calculate confidence with MaxProb. In Table 5, we show that label smoothing performs poorly in our setting. While it does not affect classifiers' ID accuracy, it significantly decreases AUROC on all but one dataset (TREC-10), where it still remains worse than CNL and all of our data generation baselines.

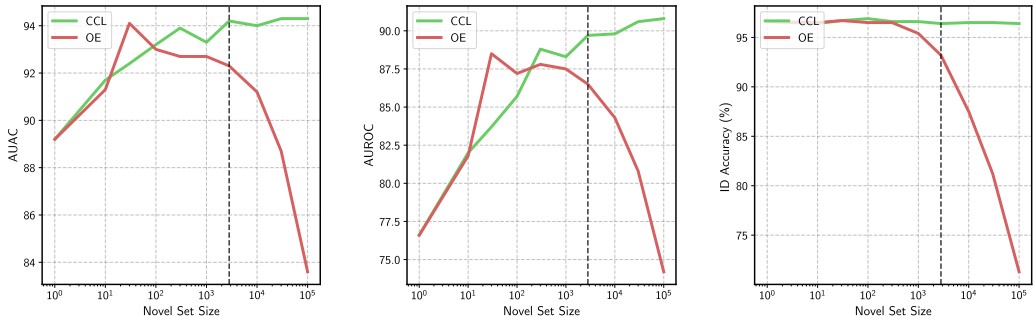

Figure 15: *Outlier Exposure is sensitive to the size of the novel set on TREC-10.* We vary the novel set size from 0 to 100K, finding that both accuracy and AUROC decrease with as few as 100 novel generations. We indicate with a dashed line the point where the novel set and training set size are approximately equal.

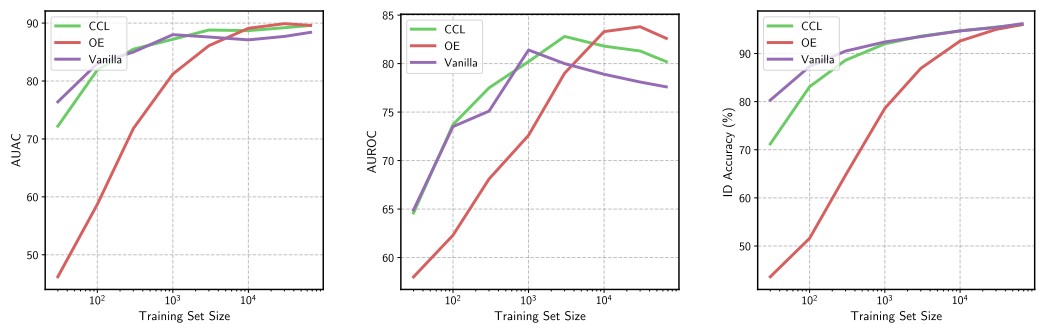

Figure 16: *Outlier Exposure disproportionately hurts smaller datasets.* We subsample the training set for AGNews, use 100K novel generated examples, and vary the training loss. We find that CCL achieves similar ID performance as Vanilla at all training set sizes, but OE significantly hurts accuracy when the training set is smaller than 1000 examples.

## A.10 OUTLIER EXPOSURE IS SENSITIVE TO GENERATED DATA

In the setting where Outlier Exposure is originally evaluated, access to some known OOD data (e.g., Wikitext) is assumed (Hendrycks et al., 2019). However in our setting, where we generate a potential novel set, there is no guarantee that the generated examples are indeed OOD. For example, we show in Appendix A.5 that less than 50% of NP generations for AGNews come from novel classes. Without this guarantee, more generated data is not always better when training with OE. One risk of using more generated novel data is that the model will see a large number of ID examples in the novel set relative to in the training set. We conduct two experiments to analyze the impact of novel set size relative to training set size.

First, we vary the novel set size relative to the training set size. In Figure 15, we train with novel sets on TREC-10 from size 0 to 100K using both OE and CCL. We observe that training with OE hurts accuracy and AUROC when the novel set is larger than 100 examples, while CCL continues to improve as the novel set size grows, and maintains accuracy for all novel set sizes. As the novel set becomes larger than the size of the training set (to the right of the dashed line), both OOD detection AUROC and ID accuracy quickly decrease. This result suggests as the ID noise the classifier sees in OE training outsizes the training set, its ID predictive ability worsens.

Of the datasets in our experiments, TREC-10 is by far the smallest, with only about 2800 training examples per split. To determine whether OE is also sensitive to the size of the ID set, we subsample the AGNews dataset into smaller training sets and perform OE and CCL training with 100K-sized novel sets. We compare the results against Vanilla training with the same ID sets in Figure 16. Although reducing the training set size decreases the ID accuracy even for vanilla training, CCL training achieves similar accuracy for all subsampling sizes. We do observe that a sub-10% accuracy

margin appears between vanilla and CCL at extremely small training set sizes, though this margin disappears at 1000 or more training examples. OE, meanwhile, decreases ID accuracy by as much as 35% when the dataset is subsampled to 30 examples, and 25%+ at 300 examples. OE-trained classifiers are also worse OOD detectors given limited training data: they underperform vanilla classifiers for all training sets smaller than 3000 examples. Finally, we find that OE does yield better OOD detectors than CCL for sufficiently large AGNews training sets. This expands on our findings in Table 2, suggesting that when there is access to a large amount of training data, in this case 10000 examples are more, OE can learn from noisy novel sets (though ID accuracy still decreases). Our results indicate that TREC-10 is not alone: As training set size becomes smaller, the ID classes becomes less well-specified, and ID examples present in the novel set induce the model to make incorrect predictions (and poor confidence estimates) on true ID test examples.

## A.11    CNL AND SEPARABILITY

To understand why CNL improves AUROC, we compare the confidence profiles of a vanilla fine-tuned classifier against those of a CNL trained classifier. Specifically, in Figure 17, we select 50 random ID examples and 50 random OOD examples from each dataset split and compute MaxProb confidences. We find that CNL decreases confidence on OOD examples, though not to the same extent on all examples. In datasets like TREC-10 and Emotion where CNL achieves stronger AUROC gains, the decrease in OOD confidence is more pronounced. Though ID test examples also decrease in confidence on all dataset splits, this decrease is less pronounced and is likely due to the confidence contrastive objective term incentivizing the model's confidence distributions to be generally less peaked.

The shifts reflected in the confidence distributions directly impact the separability of OOD and ID examples. On the Vanilla model confidence axis, it is difficult to identify a threshold above which most examples are ID and below which most examples are OOD. Given CNL confidences, OOD and ID examples are more separable. This visual separability is reflected in the OOD Detection AUROC metric.

To demonstrate the strictness of the OE objective, we plot the confidences of the same examples without (Vanilla) and with OE training in Figure 18. First, we observe that the vast majority of OOD examples have similar confidence after OE training, as they are all pushed towards minimum confidence (maximum entropy). Second, we observe that OE significantly affects the confidence of ID test examples, decreasing the confidence of some examples lower than that of OOD test examples.

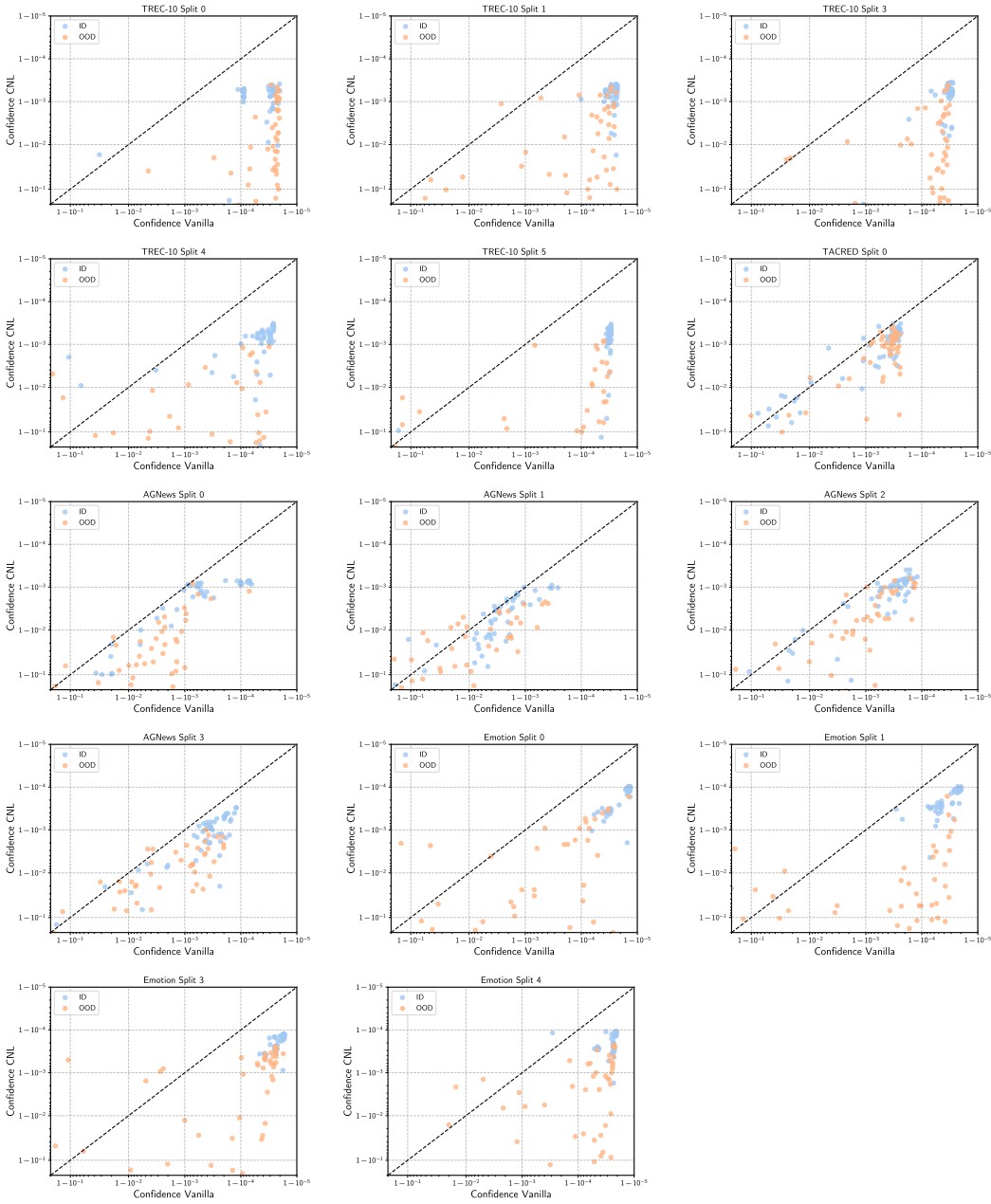

Figure 17: *CNL improves the separability of ID and OOD examples.* We plot the confidences of 50 random ID and 50 random OOD examples on a vanilla finetuned BERT classifier versus a CNL trained BERT classifier. CNL successfully decreases the confidence of OOD test examples while minimizing the impact of the confidence of ID test examples.

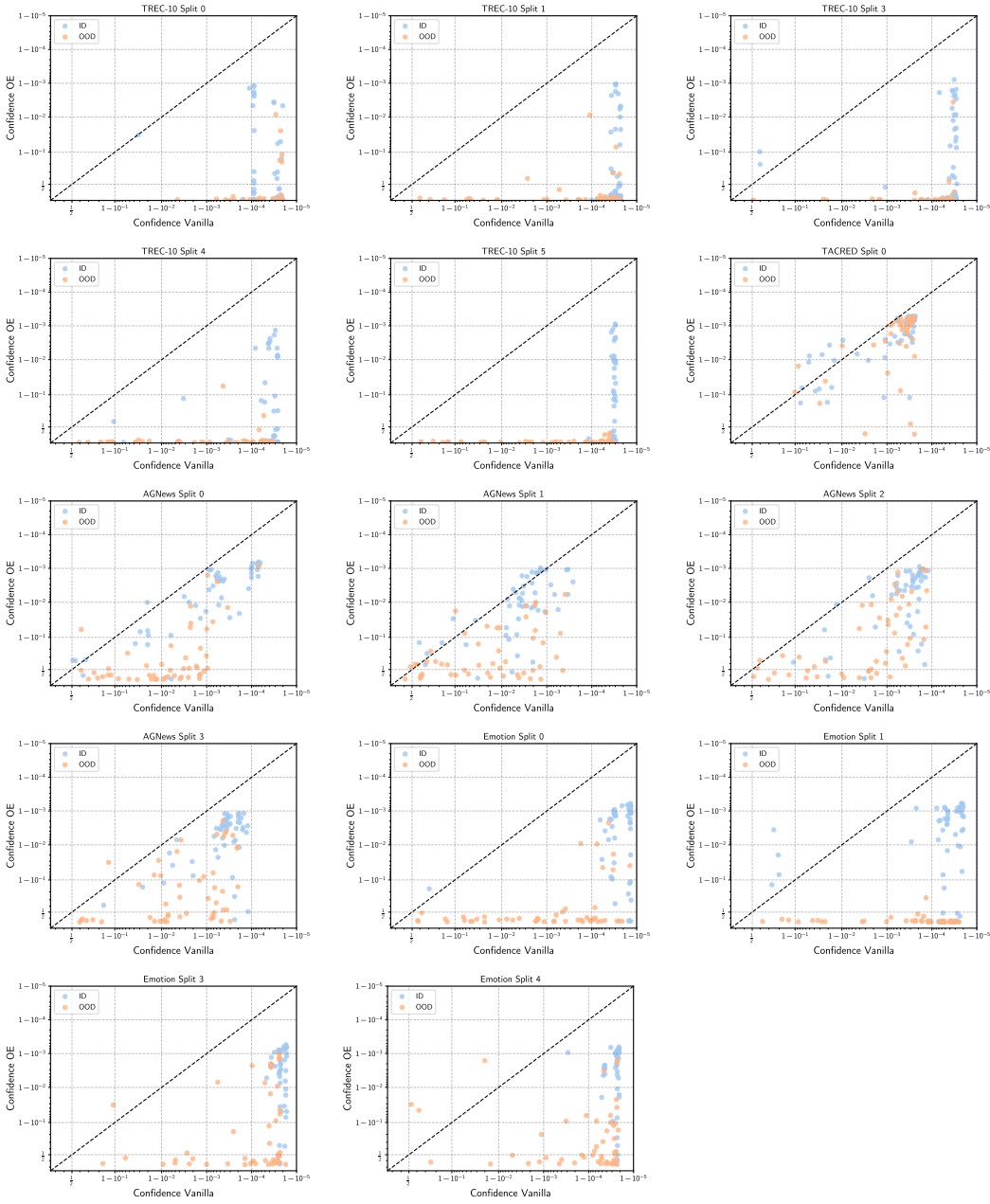

Figure 18: *OE significantly decreases the confidence of OOD examples, but unfortunately also decreases confidence on ID examples.* We plot the confidences of 50 random ID and 50 random OOD examples on a vanilla finetuned BERT classifier versus a NP+OE trained BERT classifier. We also observe that ID examples exhibit a large confidence distribution after OE training: some ID examples have similar confidence as OOD examples. Note that the axis limits on these plots differ from the axis limits on Figure 17, as confidences in general are much lower.

