# OpenReview forum: "Contrastive Novelty Learning: Anticipating Outliers with Large Language Models"
_ICLR.cc/2023/Conference — Submitted to ICLR 2023_

### Official Review · Reviewer_r6H5 · 2022-10-14

**Confidence:** 4
**Correctness:** 4
**Technical Novelty And Significance:** 3
**Empirical Novelty And Significance:** 3
**Recommendation:** 6

**Clarity, Quality, Novelty And Reproducibility:**

The paper is easy to follow, and the idea to use large pretrained generative model to generate OOD example is novel.

**Strength And Weaknesses:**

**Strength**
* The idea to use large pretrained generative model to generate OOD example is inspiring.
* The experimental results are strong.


**Weaknesses**
* While the results seems promising, I'm curious whether the choice of pretrained model (e.g. replace GPT-3 with GPT-2) impact the performance much. If the proposed method only works well on very large pretrained generative models, the computational cost could be extremely high, since the authors generate 100k OOD samples.

* The proposed method seems hard to be extended to other tasks like sequence labeling and other domains like computer vision.

**Summary Of The Paper:**

To tackle the problem of OOD detection (especially in text classification task), the authors propose a method called Contrastive Novelty Learning (CNL). The core idea is to generate OOD examples by prompting a large language model twice, first to generate label, then to generate the content. Experiments show effectiveness of the proposed method.

**Summary Of The Review:**

See Strengths/Weaknesses

---

> ### Author Response · Authors · 2022-11-17
> **Response to Review**
>
> > The idea to use large pretrained generative model to generate OOD example is inspiring.
>
> Awesome! We hope that future work can explore more diverse applications of LLM prompted generation.
>
> > While the results seems promising, I'm curious whether the choice of pretrained model (e.g. replace GPT-3 with GPT-2) impact the performance much.
>
> Great question. In the revised version, we’ve done an experiment where we replace the GPT-3 model for label generation with a smaller, open-source GPT-J language model. Performance is on-par with GPT-3 on 3 of 4 datasets, although it does seem to matter sometimes, as on AGNews the generated labels from GPT-J lead to slightly worse performance (< 1 point AUAC).
>
> > If the proposed method only works well on very large pretrained generative models, the computational cost could be extremely high, since the authors generate 100k OOD samples.
>
> First, it does not seem like very large (175B scale) generative models are necessary to generate good novel examples. Based on our experiment above, a medium sized GPT-J 6B can be used for both label and example generation and achieve competitive performance. Furthermore, in Section 5.3, we show that even CNL with smaller generative models like GPT2-large can improve OSSC over vanilla. Second, we show in Figure 6 that moderate sized novel sets (e.g., ~1000 examples) of the same or lower order of magnitude than the training set can also result in meaningful OSSC improvements (and doesn’t ****hurt**** performance for any novel set size). In practice, this means that practitioners can choose the specific generation quota that is realistic for their computational budget.
>
> > The proposed method seems hard to be extended to other tasks like sequence labeling and other domains like computer vision.
>
> Though we think this is a great question to ask, we think our method could be easily transferred to other domains. The general framework of generating and training to abstain on novel class examples can apply to any domain where we can condition on a label and generate an input (e.g, diffusion models in computer vision). Though the specific nature of the generation setup might need to change (there doesn’t seem to be a direct analogue for in-context learning in vision), future work exploring the combination of a LLM for label generation with an image input-generating model seems promising. Certainly image classification tasks with large label spaces like ImageNet can suffer from open-set shift, and this would be a great angle of approach.
>
> As for sequence labeling, we are not sure how open-set shift could manifest in this task. Do you have a specific example you were thinking of?

---

### Official Review · Reviewer_KAeh · 2022-10-21

**Confidence:** 3
**Correctness:** 3
**Technical Novelty And Significance:** 3
**Empirical Novelty And Significance:** 3
**Recommendation:** 6

**Clarity, Quality, Novelty And Reproducibility:**

see above for detail.


Typos:

- Eq 1 has a missing parenthesis.
- Table 1's caption: "Novlty"

**Strength And Weaknesses:**

The data augmentation method is straightforward/creative, and I am
glad to see that it works: it's a direct approach to ood detection
(i.e., generating ood data) -- I am a fan of things that work :-) The
paper is easy to follow, and the baselines which the authors compared
to are thorough. I appreciated the oracle experiments which frame how
much potential room for improvement there may be in follow up
work. The evaluation metrics made sense. The authors report averages
over a number of runs, yielding confidence intervals.

The biggest negative of the paper, in my view, is that the performance
gains (particularly of the new loss function) are marginal. The new
loss function outperforms OE mostly because OE seems to fail on
TREC-10 (else, it's competitive). The reason for OE failing is
"attributed to noisiness in generation" of the OOD data --- this
causes some concern, because while the new loss function seems to
overcome this noise in the other dataset cases, I would be a bit
nervous using it with any loss function given the massive performance
drop on TREC-10: is it worth the risk of involving GPT-3 for a few AUC
points? The new data augmentation, while resulting in performance
improvements over kFolden and Contrastive, also has more dependencies,
i.e., access to GPT3. In more detail, a few technical concerns:

- All of the out-of-label-distribution examples are synthetic,
  whereas, all of the in-label-distribution examples are (presumably)
  human written from the original corpus. Are models implicitly just
  doing a "human-authored" vs. "machine-authored" classification task?
  This question is partly addressed --- given that the test sets are
  human-written, the method isn't only doing machine vs. human. But,
  I'd suggest to add an additional experiment that goes the other way:
  that is, an experiment where, at test time, the
  out-of-label-distribution examples are human-written (from the
  corpora), and the in-label-distribution examples are autogenerated.

- The authors propose a new loss function in eq. 1 that goes beyond
  the uniform distribution loss of Hendrycks et al. 2019. While I can
  somewhat qualitatively see why this loss might be different than the
  uniform loss, the authors argue that it works better because it
  enables "some novel examples can be more novel than others." Can
  this claim be validated by examining the logits of the OE model
  vs. this one? The uniform loss presumably (in a soft way) also
  allows some examples to be more novel than others. The empirical
  impact of this loss function is also not entirely clear. OE seems to
  do badly on TREC-10 for some reason, but it generally seems to match
  (or sometimes outperform) the new proposed loss on the other corpora.

- I appreciated the oracle experiments --- but, somewhat
  inexplicably, OE seems to do much better with the gold data
  available vs. CCL. The reason for this is not explored deeply ---
  why is that?

- All experiments are conducted with BERT-Base, which is a 4 year old
  model at this point. While the authors take some steps to run with
  improved models like Roberta-Large, it would have been nice to see
  additional scaling runs, e.g., with larger models.

- The authors method can only work with text classification tasks,
  which is great, but --- some of the competing methods are more
  general, and the Wikitext + CCL method (which simulates the more
  general setting of not having GPT-3 but only an OOD set) only
  marginally outperforms prior solutions (kFolden, for example,
  doesn't even depend on the external set).

**Summary Of The Paper:**

The authors study out-of-distribution detection for text
classification.  They propose "contrastive novelty learning," which
combines two methods: 1) data augmentation with GPT-3 for
out-of-distribution classes; and 2) a loss function which encourages
the maximum probability scores to be lower on out-of-distribution
labels. The authors report a few accuracy points of gain compared to
prior methods, and conclude by exploring a number of questions, e.g.,
about limiting GPT-3 budget, and training with larger models.

**Summary Of The Review:**

Overall: the authors propose to do OOD-label detection by explicitly
generating OOD data with GPT3: a new loss function seems to overcome
noise in this process which causes a prior loss function to fail. The
paper is well-written, creative, and the results promising. But, I
have some doubts that the performance gains of the new method (a few
AUC points) will entice folks to undertake the suggested protocol in
practice, esp. given the GPT-3 dependence and also, the fact that
choosing a slightly different loss function can cause catastrophic
failure (e.g., on TREC-10).

---

> ### Author Response · Authors · 2022-11-17
> **Response to Review (2/2)**
>
> > I appreciated the oracle experiments --- but, somewhat inexplicably, OE seems to do much better with the gold data available vs. CCL. The reason for this is not explored deeply --- why is that?
>
> This strength of OE with gold data is a direct consequence of its strictness: when the novel set is known to be OOD in some way, it is both reasonable and desirable to minimize its confidence to the greatest degree possible. CCL on the other hand is looser: it does not take advantage of known OOD examples as well, as it only reduces confidence on these examples below that of the ID training examples.
>
> This brings up a good point. We want to make clear that do not claim that OE is a bad method for improving classifier OOD detection for known (or even representative) OOD shifts. In fact, we have good results with OE when we use an implausible dataset (Wikitext) as the novel set and even very strong results when the actual test-time distribution shift is known (known OOD novel set). However, our work considers a different setting where there is no information about the distribution shift. Because we generate potential novel examples (a process which inherently admits some noise), OE is not a good fit for our setting and we propose the CCL alternative.
>
> > it would have been nice to see additional scaling runs, e.g., with larger models.
>
> In the revised version, we have included additional experiments with two new models, DeBERTa-v3-base and DeBERTa-v3-large. We find that CNL improves OSSC on average for both models. Another observation is that vanilla finetuning for both DeBERTa-v3 models does not result in better AUAC than RoBERTa, even though the ID accuracy is slightly higher. Though it’s not clear what the specific reason is, it does suggest that newer/better models in terms of accuracy (or even OOD generalization) may not be better at *recognizing* OOD examples, as they are not explicitly trained for this task.
>
> > The authors method can only work with text classification tasks, which is great, but --- some of the competing methods are more general
>
> Does this comment refer to the focus on NLP tasks, or on classification?
>
> For the first point, though we only consider NLP tasks for now, the general framework of generating and training to abstain on novel class examples can apply to any domain where we can condition on a label and generate an input (e.g, diffusion models in computer vision). In fact, it might be very interesting to pair LLM-based label generation with image generation, as it’s hard to reproduce purely ICL-reliant methods like the few-shot baseline.
>
> As for the second point, we only consider classification tasks because open-set shift is well-defined in this sequence. In sequence-to-sequence tasks like question answering, for example, it’s not very clear how the output space could change in a way that is not accounted for during training, as there are usually no strong assumptions made at training time about the output space.
>
> > the Wikitext + CCL method (which simulates the more general setting of not having GPT-3 but only an OOD set) only marginally outperforms prior solutions (kFolden, for example, doesn't even depend on the external set).
>
> We don’t think that the weak performance of Wikitext + CCL is concerning or surprising. The motivation of our work is to anticipate outliers with a LLM and train for abstention on these outliers. Wikitext + CCL does not adhere to that goal, and is simply an analogue for what is done in Hendrycks et al. 2019, where we train for abstention on an arbitrarily chosen external dataset. Instead, the 5+ point AUROC gain from Wikitext + CCL to CNL (NP + CCL) indicates that generating these outliers is very important for improving the classifier’s ability to abstain on novel class examples.
>
> As for dependence on GPT-3, we show in Appendix A.3 and Figure 12 that we can remove GPT-3 from the system entirely and still maintain performance on 3 of 4 datasets.
>
> > Typos
>
> Thanks for being so perceptive! We’ve fixed both these typos.
>
> (Also, don't mind the deleted comment below. It was reposted so that these could appear in the right order.)

---

> > ### Comment · Reviewer_KAeh · 2022-11-29
> > **Thanks for the thorough reply!**
> >
> > Hi There, Just acknowledging that I saw+read your reply --- thanks for the effort+care you took in responding. I will take this into account during reviewer discussions, which you might not be able to see but are occurring.

---

> ### Author Response · Authors · 2022-11-17
> **Response to Review (1/2)**
>
> > I am a fan of things that work :-)
>
> Great! As are we.
>
> > while the new loss function seems to overcome this noise in the other dataset cases, I would be a bit nervous using it with any loss function given the massive performance drop on TREC-10: is it worth the risk of involving GPT-3 for a few AUC points?
>
> We believe that the new experiments in Section 5.1 and Appendix A.10 shed more light on why this OE TREC-10 performance drop occurs (generation noise), and simultaneously show that CCL is robust to to generation noise by design. Henceforth we refer to this accuracy drop due to generation noise as “sensitivity.” Specifically, in Figure 3 we observe that in a controlled setting, CCL is not sensitive to noising even as the noise percentage nears 100%. The same is not true for OE, which drops in accuracy. We find similar insensitivity to novel set size for CCL in Figure 15. In Figure 16, there does exist an accuracy drop in training set sizes of between 30 and 300 examples for CCL, though in this regime the vanilla finetuned model also achieves significantly worse accuracy, indicating that the ID class decision boundaries are not well-enough specified by the limited amount of training data.
>
> > I'd suggest to add an additional experiment…at test time, the out-of-label-distribution examples are human-written (from the corpora), and the in-label-distribution examples are autogenerated.
>
> This indeed sounds like a very interesting experiment. Given time constraints, we were not able to complete this experiment yet, but will consider this for the next version.
>
> > The new loss function outperforms OE mostly because OE seems to fail on TREC-10 (else, it's competitive). The reason for OE failing is "attributed to noisiness in generation" of the OOD data
>
> We’ve added several experiments to understand why OE suffers ID accuracy drops on TREC-10. The general response summarizes our findings in greater detail, but the most direct finding is that TREC-10 has only about 2800 training examples in each split, while the next smallest dataset, Emotion, has about 8000+ examples. As we use a large novel set of 100K examples, the model will be trained to maximize entropy on many ID examples in the novel set, hurting performance. In Figure 15, we see that ID accuracy steeply drops as the novel set is much larger than the train set. This would certainly be less severe for other datasets, which are at least several times larger. OE also hurts accuracy on AGNews and Emotion, as shown in Table 3, which, although lower, is still concerning/undesirable. Given these observations, we believe that the only reasonable solution is to design a loss tolerant to generation noise, as affecting the ID performance in any substantial way is bad.
>
> > The uniform loss presumably (in a soft way) also allows some examples to be more novel than others. The empirical impact of this loss function is also not entirely clear.
>
> Though OE is not completely strict in the sense that it requires all novel set examples to have completely uniform output distribution, it is a stricter objective than CCL, which is evident when ID noise is present in the novel set (see Figure 3). The nice aspect of CCL is that it compares the confidence of OOD examples against ID examples, allowing ID examples in the novel set to have equal confidence to ID examples in the train set and OOD examples in the novel set to have relatively lower confidence. To demonstrate this effect, we’ve added some plots on the MaxProb confidences of the classifier on random test examples with and without CCL training in Appendix A.11 and Figures 17 and 18.
>
> We make two observations from looking at how the CCL confidence landscape changes (Fig. 17):
>
> 1. **Confidence Change:** While OOD examples tend to have similar/high confidence as ID examples without CCL training, they almost all decrease in confidence with CCL training.
> 2. **Separability:** On the y-axis (with CCL) the OOD and ID examples are very separable, whereas on the x-axis it’s visually difficult to draw a decision boundary between OOD and ID examples.
>
> In contrast to CCL, we observe a different outcome in Fig. 18 regarding OE:
>
> 1. **OOD examples:** The vast majority of OOD examples have similar confidence after OE training, as they are all pushed towards minimum confidence (maximum entropy). This is evidence for our claim that OE is “stricter.”
> 2. **ID examples:** OE significantly affects the confidence of ID test examples, decreasing the confidence of some examples lower than that of OOD test examples.
>
> We hope that these plots lend insight on how OE/CCL training differs. Thank you for noting this!

---

### Official Review · Reviewer_i6wa · 2022-10-24

**Confidence:** 3
**Correctness:** 2
**Technical Novelty And Significance:** 2
**Empirical Novelty And Significance:** 2
**Recommendation:** 5

**Clarity, Quality, Novelty And Reproducibility:**

Clarity:
some important details are missing.

Novelty:
It brings some new ideas to OOD detection and classification.

Reproducibility:
Others can reproduce the results.

**Strength And Weaknesses:**

Strength:
Generating OOD examples with huge PLMs is an interesting idea.

Weaknesses:
Some important details are missing.
Improvements over the baselines are not fully convincing.

**Summary Of The Paper:**

The paper deals with OOD examples in classification. It leverages a huge PLM and generates a few examples that distribute differently from the ID examples. Subsequently, it learns a classifier with a contrastive confidence loss. Experimental results on a few public datasets indicate the effectiveness of the proposed method on classification and OOD detection.

**Summary Of The Review:**

The paper attempts to train a classifier that can accurately detect OOD examples. The interesting part is novelty prompting in which examples that are semantically novel and bear some resemblance to ID examples are generated with prompts using huge language models. However, I have some concerns regarding to the presentation and the empirical results:
(1) CCL+few shot data augmentation: the description is "prompt with only an instruction and zero or more ID training examples", then how many ID training examples are involved in CCL+few shot? What is the difference between the proposed method?
(2) what is the classification model?
(3) Looking at Table 1, I am curious why OE+novelty prompting performs such badly on TREC-10. If it is sensitive to noise, then why it renders descent results on the other three datasets? This is critical, as the baseline outperforms the proposed method on two of the four datasets but loses finally on the "average score".
(4) the improvement over CCL+few-shot is marginal. That's why I am curious about the difference between the proposed method and CCL+few-shot.

---

> ### Author Response · Authors · 2022-11-17
> **Response to Review**
>
> > CCL+few shot data augmentation: the description is "prompt with only an instruction and zero or more ID training examples", then how many ID training examples are involved in CCL+few shot? What is the difference between the proposed method?
>
> This baseline is meant to be an ablation of the full CNL method we present. We remove the label generation step and prompt only with one example from each class (the same number of in-context examples as our regular example generation step). The specific number of in-context examples is 3-6 for our datasets, depending on the number of classes in the training set. Thank you for your observation. We’ve directly explained this in the revised version.
>
> > what is the classification model?
>
> Though this was previously stated in Section 4.3, we have also mentioned in Section 4.2 to improve clarity. We use BERT-base as the classifier model for our main experiments, but in Section 5.3 we show that CNL can also improve the OSSC ability of other classifiers like RoBERTa and DeBERTa-v3, of both base and large sizes.
>
> > Looking at Table 1, I am curious why OE+novelty prompting performs such badly on TREC-10. If it is sensitive to noise, then why it renders descent results on the other three datasets? This is critical, as the baseline outperforms the proposed method on two of the four datasets but loses finally on the "average score".
>
> This is a great question, and the experiments we did to investigate this sensitivity (in both Section 5.1, Figure 3 and Appendix A.10, Figures 15 and 16) led to some very interesting results. The most direct explanation why TREC-10 suffers the greatest accuracy drop is that it is the smallest dataset we consider: it has only about 2800 training examples in each split, while the next smallest dataset, Emotion, has about 8000+ examples. As we use a large novel set of 100K examples, the model will be trained to maximize entropy on many ID examples in the novel set, hurting performance. In Figure 15, we see that ID accuracy steeply drops as the novel set is much larger than the train set. This would certainly be less severe for other datasets, which are at least several times larger.
>
> So why not use a smaller novel set paired with OE? We’ll mirror an argument made to reviewer b3bf here: Though there appears to be an optimal inflection point of OOD set size for OE in Figure 15, finding this inflection point is usually not possible.  First, given a generated novel set, such an optimum may not be very good: even if you generate a novel set of only 10 examples, there is some risk that several examples will be ID, and the smaller this generated set is, the less likely it is to account for many possible test-time novel classes. For example, in our experiment, we find that novel set size N=300 is a good value for TREC-10. However, the AUAC and AUROC is still less at N=300 than for CCL with N=100K. Second, in our setting, evaluating on the true open-set class is impossible, as the nature of the distribution shift is assumed to be unknown.
>
> As for why OE performs well on other datasets, it seems based on Table 3 that ID accuracy drops also occur on AGNews (96.1 → 95.6) and Emotion (97.7 → 96.4), suggesting that all is not well there either. With our AGNews subsampling experiments in Figure 16, we show TREC-10 is not unique, but that the OE ID accuracy drop is more apparent when the train set is smaller. Thus OE is still sensitive to generation noise, it is just less apparent due to access to a large training set.
>
> > the improvement over CCL+few-shot is marginal. That's why I am curious about the difference between the proposed method and CCL+few-shot.
>
> In Table 2, we show that Novelty Prompting-based generation does improve significantly in abstention on OOD examples over CCL+Few-shot on 3 of 4 datasets. We show that In Appendix A.5., Figure 14 that these improvements are due to NP generating more novel class examples than Few-shot. Finally, CCL+Few-shot is simply an ablation of our presented method (CNL), and we hope that its inclusion improves understanding of what factors make our method successful. Specifically, our main motivation is to automatically generate novel class examples using a large-language model. Instead, Few-shot generation’s surprisingly strong performance demonstrates that pretrained language models are unexpectedly good at inferring that the generated example should differ in label from the in-context examples without label supervision, though can still benefit from our label generation guidance.

---

> ### Author Response · Authors · 2022-12-01
> **Quick Follow-up**
>
> Hello! Let us know if you have any additional questions, especially regarding clarity, any details you think might be missing, and any remaining aspects of the prior response. We would be happy to provide additional clarification before the end of the rebuttal phase.

---

### Official Review · Reviewer_b3bf · 2022-10-26

**Confidence:** 2
**Correctness:** 3
**Technical Novelty And Significance:** 3
**Empirical Novelty And Significance:** 2
**Recommendation:** 6

**Clarity, Quality, Novelty And Reproducibility:**

The overall clarity and quality of the work is good. Use of GPT3 might limit reproducibility but it won't be a permanent problem.

**Strength And Weaknesses:**

Strength

- The idea is well motivated, both wrt the novelty prompt generation and the contrastive confidence loss.
- Empirical results are overall positive, with reasonable baselines and comparisons and some ablation studies.
- The writing is clear and easy to follow.

Weaknesses

- There are some details that are not very clear
  - S4.3 For contrastive baseline, I'm not sure how to interpret "pairwise distant but internally compact" in the context here. Are there novel classes and examples involved? Might be good to clarify this question for the baselines here?
  - S5.1 it was a bit difficult to follow the various comparisons made in ref to Table 1. E.g., I had a hard time figuring out what is being compared at "This accuracy decrease is smaller"
  - S5.2 "To isolate the factors..." what are the factors being isolated here and how is the isolation achieved by Table 2?
  - S5.3 Generator size is concerned only with the example generator, while the novel-label generator is always GPT3? What happens if one generates novel labels with smaller models?
- There's not much error analysis other than offhand comments with limited support.
  - For example, when discussing vanilla outperforming OE+NP ("We attribute this behavior to the noisiness of generation: some novelty prompted examples are similar to ID examples"), although I intuitively agree with this sentiment, it would be great to have some empirical support by quantifying example similarity.

**Summary Of The Paper:**

The paper proposed a two-step generation-based method for open set selective classification. The generation process starts with first generating a set of OOD labels then examples conditioned on these novel labels. The paper also proposed a new training objective that pushes the max probability of ID predictions above those of OOD predictions, which is less restrictive than existing proposals by avoiding global uniformity. The proposals are evaluated on a few classification tasks on NLP and empirically shown to be effective, with ablation on some selected aspects of the setup.

**Summary Of The Review:**

The paper proposed a reasonable solution to an important problem with good empirical results and decent result analysis. Publication would benefit the community.

---

> ### Author Response · Authors · 2022-11-17
> **Response to Review**
>
> > The idea is well motivated, both wrt the novelty prompt generation and the contrastive confidence loss.
>
> Thanks!
>
> > S4.3 For contrastive baseline, I'm not sure how to interpret "pairwise distant but internally compact" in the context here…
>
> We’ve corrected the writing to be more direct with the explanation of the contrastive baseline. The intended explanation is that the contrastive objective induces same-class examples to be closer and different-class examples to be further in representation space, leading to well-separated clusters of examples which in theory should lie further from novel class test examples.
>
> We also included a sentence that summarizes which methods use novel classes/examples (only OE and CCL).
>
> > S5.1 it was a bit difficult to follow the various comparisons made in ref to Table 1. E.g., I had a hard time figuring out what is being compared at "This accuracy decrease is smaller"
>
> The accuracy numbers referred to in the paper are contained in the appendix. To minimize the need to refer to those numbers, we have included numbers discussed in comparisons directly in the main text.
>
> As for the specific comparison above, this comment made us realize that we had copied an erroneous evaluation number in Table 3 (OE + Gold, TREC-10). We have corrected it in the revised version and removed the reference to the old number in section 5.1.
>
> > S5.2 "To isolate the factors..." what are the factors being isolated here and how is the isolation achieved by Table 2?
>
> As AUAC is a holistic metric, improvement can result from one of three possibilities. The first (and least likely reason) is that the ID accuracy increases. The other two, improving OOD abstention and improving abstention on ID model errors, are both possible reasons for our improvement. To confirm that we do in fact achieve our stated goal (improve abstention on open-set examples), we measure the AUROC on the OOD detection (i.e., abstention) task. We have updated this to instead read “To confirm that CNL improves a classifier’s ability to distinguish novel class examples.”
>
> > S5.3 Generator size is concerned only with the example generator, while the novel-label generator is always GPT3? What happens if one generates novel labels with smaller models?
>
> Thanks for the suggestion! We’ve included an additional experiment where we use GPT-J as both the label and example generator in Appendix A.3. In Figure 12, we show that GPT-J performs on-par with GPT-3 label generation on 3 of 4 datasets. This indicates that our method is not dependent on GPT-3, but can be applied with different choices of generators.
>
> > There's not much error analysis other than offhand comments with limited support.
>
> Because we introduce a method composed of multiple intermediate steps, we agree that it is important to understand where our method performs well and which steps admit errors that can be improved upon in future work. First, we perform manual annotation of the labels generated from our method. We show that label generation is not perfect, as in some datasets 50%+ of generated labels can be closed-set (ID), but that in most cases less than 15% of generated labels are ID.  Our prior manual analysis focuses on the following phase of our method example generation, through annotation of generated AGNews examples. Finally, we plot the confidence of test examples before and after CNL training, where we observe that CNL training benefits OOD detection because it improves the confidence-based separability between ID and OOD examples.
>
> > when discussing vanilla outperforming OE+NP…it would be great to have some empirical support
>
> We’ve included three new experiments in Section 5.1 and Appendix A.10 demonstrating that (1) in a controlled setting, generation noise hurts ID accuracy for OE but not CCL, (2) when the novel set is much larger than the training set, this ID accuracy drop is more severe, which explains the issues with TREC-10, and (3) this can apply to other datasets, namely AGNews, when the training set becomes smaller. Refer to the main response for further explanation.
>
> *Does that mean we can simply use a smaller OOD set in combination with OE?* Though there appears to be an optimal inflection point of OOD set size for OE, finding this inflection point is usually not possible.  First, given a generated novel set, such an optimum may not be very good: even if you generate a novel set of only 10 examples, there is some risk that several examples will be ID, and the smaller this generated set is, the less likely it is to account for many possible test-time novel classes. For example, in our experiment, we find that novel set size N=300 is a good value for TREC-10. However, the AUAC and AUROC is still less at N=300 than for CCL with N=100K. Second, in our setting, evaluating on the true open-set class is impossible, as the nature of the distribution shift is assumed to be unknown.

---

### Author Response · Authors · 2022-11-17
**General Response**

Dear Reviewers,

Thank you for all the thoughtful feedback! We appreciate that reviewers found our method well-motivated and creative, and the writing easy to follow. We’ve made a number of improvements based on feedback and included additional experiments to address reviewer concerns. As a summary of the major updates, we:

1. Performed an experiment where we use only GPT-J models for both label generation and example generation in Appendix A.3. We find that this performs as well as our main method (GPT-3+GPT-J) on 3 of 4 datasets, and within 1% AUAC on the last dataset.
2. Investigated the mismatch of Outlier Exposure (OE) with novelty prompted data through three analysis experiments in Section 5.1 and Appendix A.10. In the process, we show that TREC-10 is not the only setting where OE underperforms: instead, OE is very sensitive to ID noise in generation and the size of the novel set relative to the training set, making it poorly suited for use with generated data.
3. Performed several error analysis experiments on the intermediate generations and final trained classifiers in Appendices A.5 and A.11. We annotate the labels automatically generated from our label generation step to understand the remaining obstacles for novel example generation. We also visualize classifier confidence on test examples before and after CNL training.
4. Extended the classifier model ablations in Section 5.3 to newer models, specifically DeBERTa-v3 (base+large)
5. Made several writing updates, including additional details for our method and baselines.
6. Fixed a single erroneous evaluation number in Table 3, Appendix A.2 (OE + Gold Data, TREC-10).

**GPT-J as a Label Generator**

Several reviewers expressed concerns that usage of GPT-3 would hinder adoption of this method. We agree, and test whether a smaller, open-source model can do label generation. We perform an experiment use GPT-J 6B for both label and example generation, removing all reliance on the GPT-3 API. We find that CNL with GPT-J only performs as well as our prior GPT-3/GPT-J method on 3 of 4 datasets, and within 1% AUAC on the remaining dataset. We discuss this result in Appendix A.3 and Figure 12.

**Analyzing OE’s Performance with Generated Data**

Another concern is that training with OE seems to hurt ID accuracy significantly more on TREC-10 than the other three datasets. We also found this to be a curious result, and have interesting findings to share that support claims we made in the last version. We performed three experiments:

1. **OE does not smoothly handle generation noise:** In Section 5.1 and Figure 3, we analyze the impact of generation noise in the novel set on in-distribution accuracy for both OE and our loss (CCL). In this setting, we define problematic generation noise as the unintended generation of closed-set class (ID) examples in our purported novel set. This pitfall is almost inevitable when generating OOD examples fully automatically. We take a set of gold OOD examples and noise it with various proportions of heldout ID examples, producing novel sets of various noise levels. Training against these noised novel sets, we find that OE training leads to accuracy decreases at 30% or more ID noising, while CCL does not significantly hurt accuracy at any noise level. *This experiment clarifies the primary reason why we consider CCL: OE is hard to pair with  imperfect OOD data.* So why is OE so bad on TREC-10?
2. **OE Accuracy drops on TREC-10 are larger when using larger novel sets**: We theorize that the more ID noise the classifier sees in OE training, the worse its ID predictive ability. With a larger novel set, the expected amount of ID noise seen increases. In Appendix A.10, we find this to be true: as shown in Figure 15, when the novel set (100K examples)  is significantly larger than the training set (2.8K examples), accuracy decreases drastically. We generally find a negative correlation between the novel set size and ID accuracy with OE training. In contrast, CCL maintains ID accuracy at all novel set sizes.
3. **Smaller datasets suffer more from noisy generation+OE**: TREC-10 is by far the smallest dataset we consider, making it easy for generation noise to overwhelm signal from the train set. To test whether OE noise sensitivity applies on other datasets, we consider a smaller training set from another dataset. In the second experiment of Appendix A.10, we take subsample AGNews training sets to smaller sizes, ranging from 30 to the full ~67K examples and train with 100K novel examples. In Figure 16, we find that as the training set becomes smaller, the ID accuracy gap between OE and Vanilla training increases, going up to 35%+ acc gap at 30 ID examples. Meanwhile the ID accuracy gap between CCL and Vanilla is less than 10% even at small training set sizes. Our finding here is that TREC-10 is not unique — OE can suffer from generation noise on other datasets when the training set size is not large enough.

---

### Decision · Program_Chairs · 2023-01-20

**Decision:**

Reject

**Justification For Why Not Higher Score:**

Please see above.  I don't think there is not enough enthusiasm regarding the paper both amongst the reviewers and myself.

**Justification For Why Not Lower Score:**

N/A

**Metareview: Summary, Strengths And Weaknesses:**

This paper presents a method called Contrastive Novelty Learning (CNL), a two step method that generates out of distribution examples representative of novel classes, then trains another model to decrease confidence on them.  The authors present results that produce classifiers that that improve their ability to detect an abstain from classes compared to prior method by 2.3% AUAC and 5.5%
AUROC across 4 text classification datasets.

Strengths:  Both the reviewers and I found the paper to be well written, the results to be positive, and the method is quite straightforward.

Weaknesses:  I did not find the paper to be extraordinarily novel as synthetic data generation of a certain kind (representing a certain phenomenon) and then training smaller models towards improving on those phenomena is not very novel;  text classification itself is not a very interesting NLP task.  Generally, due to this observation and also the results being somewhat unconvincing to some reviewers ("improvements are marginal", the overall response from the reviewers is not overwhelmingly positive.  Some reviewers also demand more clarity in comparisons with baselines, and comment regarding the weaknesses in the experiments section.  Finally, there are questions about more tasks in NLP beyond classification--I agree with this sentiment as I don't find text classification is not a very inspiring problem to focus on.  Working on a few more families of NLP problems would have made the paper stronger.